# Major improvement of altimetry sea level estimations using pressure derived corrections based on ERA-Interim atmospheric reanalysis

L. Carrere[1], Y. Faugère[1]  and M. Ablain[1]

[1]Collecte Localisation Satellites, Parc Technologique du Canal, 8-10 rue Hermès, 31520 Ramonville-Saint-Agne, France

*Correspondence to*: L. Carrere (lcarrere@cls.fr)

**Abstract.** New Dynamic Atmospheric Correction (DAC) and Dry Tropospheric (DT) correction derived from the ERA-Interim meteorological reanalysis have been computed on the 1992-2013 altimeter period. Using these new corrections improves significantly sea-level estimations for short temporal signals (< 2 months); the impact is stronger if considering old altimeter missions (ERS-1, ERS-2, Topex/Poseidon), for which DAC_ERA allows reducing the along-track SSH error by more than  3 cm in the Southern Ocean and in some shallow water regions. The impact of DT_ERA is also significant in the southern high latitudes for these missions.

Concerning more recent missions (Jason-1, Jason-2, and ENVISAT), results are very similar between ERA-Interim and ECMWF based corrections: on average on global ocean, the operational DAC becomes slightly better than DAC_ERA only from year 2006, likely due to the switch of the operational forcing to a higher spatial resolution. At regional scale, both DACs are similar in deep ocean but DAC_ERA raises the residual crossovers variance in some shallow water regions indicating a slight degradation on the most recent years of the study. On the second decade of altimetry, unexpectedly DT_ERA still gives better results compared to the operational DT.

Concerning climate signals, both DAC_ERA and DT_ERA have a low impact on global MSL trend, but they can have a strong impact on long-term regional trends estimation, until several mm/yr locally.

## 1. Introduction

Since the 1990's, several altimeter missions have been monitoring the sea level at global scale. Thanks to its current accuracy and maturity, altimetry is now considered as a fully operational and accurate observing system dedicated to scientific and operational applications, among which understanding the global climate change and the related global Mean Sea Level rise (MSL) and mesoscale applications are a priority.

Satellite altimetry has shown its efficiency to detect early changes in the global and regional MSL trends (Willis and Church, 2012; Cazenave et al., 2014). However ensuring the long-term consistency and stability of altimeter measurements from one or several missions is challenging.

The global MSL trend has been determined to be around 3.2 mm/yr over the period 1993-2008, with an uncertainty of 0.5 mm/yr (Ablain et al., 2009, 2015) mostly explained by the orbit errors (Couhert et al., 2014), the ageing of the altimeters'

instruments, the drifts detected in radiometer wet tropospheric correction (Legeais et al., 2014) and uncertainties due to geophysical corrections.

In order to access the targeted ocean signal, altimeter measurements are corrected from several instrumental and geophysical corrections including the Dry Tropospheric correction (noted DT), and the Dynamic Atmospheric Correction (DAC) which

is one of the most critical after the tide correction.

The accuracy of DAC has been deeply improved during the last 20 years. First, because the ocean has a clear dynamic response to atmospheric forcing at high frequencies and when considering large scales (Vinogradova and Ponte, 2007; Mathers and Woodworth, 2001;Ponte and Gaspar, 1999; Willebrand et al., 1980), taking into account a DAC instead of an static Inverse Barometer correction (noted IB) allowed a very significant improvement of altimetry product (Carrere and

Lyard, 2003). Then the quality of the DAC has increased thanks to a better bathymetry field and a higher resolution mesh from 2007 (Carrere et al., 2007). Still, significant errors remain mostly due to a lack of resolution of the model (in shelf seas but also in some deep ocean regions), to remaining bathymetry errors and also due to atmospheric forcing fields uncertainties (Lamouroux et al. 2006; Lamouroux, 2006; Greenberg et al., 2007).

In this context, the main objective of the sea-level CCI project (Ablain et al, 2015), was to build improved long-term

altimeter sea level data records dedicated to climate studies. For that purpose several algorithms (instrumental parameters, orbit calculation, radiometer wet tropospheric correction, atmospheric corrections derived from model, oceanic tidal corrections, sea-state bias, ...) were developed to improve altimetry data and the processing to merge altimeter missions together.

Concerning the pressure derived corrections DAC and DT, one of the main issues comes from the fact that the ECMWF

operational analysis used to force the barotropic model, are not compliant with climate and MSL applications. The stability is not ensured because many jumps exist in the meteorological temporal series due to ECMWF model evolutions or upgrades: Ablain et al. (2009) showed a significant impact of these jumps on the trends of the IB and the Dry Troposphere corrections which both depend on the atmospheric pressure field. Moreover the quality of the operational meteorological dataset is not homogeneous on the entire altimeter period: early years are less accurate because of the use of old versions of

the analysis system (ECMWF) and this may impact the estimation of mesoscale signals on the oldest years (Carrere, 2003).

The methodology adopted is to use the ERA-Interim meteorological reanalysis (Dee et al., 2011) to compute the new DAC_ERA and DT_ERA corrections and analyse their impact on sea level estimation at climate scales, but also at lower temporal scales for mesoscale applications. The main advantage of using meteorological reanalysis is the homogeneous quality of the temporal series, but at the cost of a lower spatial resolution.

After a complete description of the datasets and the methods of comparison in section 2, we present an analysis of the differences of the atmospheric pressure derived corrections in section 3, and the impact of the new corrections DAC_ERA and DT_ERA on ocean short scale signals in section 4. Section 5 is dedicated to ocean long-term climate signals and section 6 gathers the discussion and concluding remarks.

## 2. Description of the datasets and method

### 2.1 Altimeter data

The altimeter measurements used were produced by Ssalto/*Duacs* and are distributed by *AVISO (*Archiving, Validation, Interpretation of Satellite Oceanographic data, 2013*)*, with support from *CNES* (http://www.aviso.altimetry.fr/en/data/products/sea-surface-height-products/global.html). Particularly we have considered level-2 altimetric products, with 1Hz along-track resolution, usually called Geophysical Data Records (GDRs).

The altimeter period (from 1993) is sampled by six altimeter missions available on two different long-term tracks: TOPEX/Poseidon (noted TP in the text and on figures), Jason-1 (noted J1 on figures) and Jason-2 (noted J2 on figures), which are the reference missions flying on the reference TP track with a 10-day cycle; and ERS-1 (noted E1 on figures), ERS-2 (noted E2 on figures), ENVISAT (noted EN on figures), which fly on a sun-synchronous orbit with 35-day cycle. The cycles used for the present study are listed in .

The different missions have been homogenized (AVISO 2012, Ablain et al. 2015) and the temporal series of TP, Jason-1, Jason-2 on one hand, and ERS-1, ERS-2 and ENVISAT on the other hand, have been concatenated to produce two long-term altimeter time series as described on Figure 1. Nearly 20 years of data for each different orbit have been used for the present study, from 1993 onwards.

The altimeter Sea Surface Height (SSH) is defined as the difference between Orbit and Range, corrected from several instrumental and geophysical corrections:

$$SSH = Orbit - Range - \textbf{DAC} - \textbf{DT} - Tide - Other\_Corr \qquad (1)$$

where:

- *DAC* is the Dynamic Atmospheric Correction studied in this paper

- *DT* is the Dry Tropospheric Correction also studied in this paper

- Tide includes de the geocentric tide, the solid Earth tide and the pole tide corrections. The geocentric tide correction comes from GOT4.7 model (Ray 1999).

- Other_Corr includes the Wet Tropospheric correction, the Ionospheric correction, the Sea State Bias correction and complementary instrumental corrections if needed.

The Sea Level Anomaly (SLA) is defined by the difference between SSH and a mean profile (MP) for repetitive orbits or a Mean Sea Surface (MSS) for drifting or new orbits. Mean profiles computed on the reference period of 7-years (1993-1999) respectively for TP-Jason and ERS-ENVISAT orbits, have been used for the present study (Hernandez and Schaeffer, 2001).

## 2.2 ERA-Interim dataset

The ERA Interim meteorological dataset is the latest global atmospheric reanalysis produced by the European Centre for Medium-Range Weather Forecasts (ECMWF). Nearly thirty-four years of data (from 01/01/1979) are available on the N128 gaussian grid (equivalent to ~0.7°), which is the native resolution chosen for the reanalysis. More details about the configuration and the performances of the system are given in the ERA-Interim reanalysis report (Dee et al., 2011). When compared to the ECMWF operational analysis, ERA Interim benefits from a constant resolution and a constant model version which makes it very useful for climate studies in particular. ERA-Interim resolution is better than the operational one on the first years of altimetry (0.7° instead of 1°). Six-hourly ERA-Interim analysis grids of sea level pressure and 10 meters wind speeds have been used for the study.

## 2.3 The Dynamic Atmospheric Correction

The high-frequency (noted HF) ocean signal forced by the atmosphere has a strong variability and is mostly located at high latitudes and in shallow water regions (Willebrand et al. 1980; Mathers, 2000); it is mostly barotropic if considering large spatial scales (Vinogradova et al. 2007). This HF signal is aliased into the lower frequency band due to the bad temporal sampling of satellite altimeters (time revisit of 10 days for TP-Jason altimeters); if not corrected, this signal thus pollutes ocean circulation estimations from altimetry for mesoscale or climate applications and also for satellite calibration campaigns. This HF ocean variability thus needs to be corrected from an independent geophysical correction with centimetric accuracy (Stammer et al. 2000).

Since 2004 the Dynamic Atmospheric Correction (noted DAC) is used in altimeter GDRs; it is a combination of the high frequencies of MOG2D-G barotropic model forced by pressure and wind (Carrere 2003) and the low frequencies (noted LF) of the Inverted Barometer, assuming a static response of the ocean to atmospheric forcing for low frequencies. The filtering wavelength is based on TP/Jason-1/Jason-2 Nyquist frequency of 20 days (twice a cycle length), because this correction is primarily a de-aliasing correction made for reference altimeter missions (Carrere and Lyard, 2003).

$$DAC = MOG2D\text{-}G_{HF\,(T \leq 20\,days)} + IB_{LF\,(T > 20\,days)} \qquad (1)$$

As far as ERS and ENVISAT missions are concerned, the sampling Nyquist period is 70 days which means that the DAC does not remove all atmospheric forced high frequency signals aliased in the data. For altimeter multi-missions products (AVISO, 2011), remaining aliased signals are smoothed thanks to a long wavelength error correction; however for mono-

mission products like GDRs, these signals remain aliased in lower frequency signals and can interfere with climate/seasonal variability (Carrere et al., 2010).

The reference DAC correction is computed from the 6-hours ECMWF operational analysis (sea level pressure and 10 meters winds) as done in CNES/AVISO dataset (AVISO, 2011; Carrere and Lyard 2003). The reference DAC is noted DAC_ECMWF hereafter.

### 2.3.1 Processing of S1 and S2 atmospheric tides

As far the Dynamic Atmospheric Correction for altimetry is concerned, the diurnal (S1) and semi-diurnal (S2) atmospheric tides demand a specific processing because they generate radiationnal tides at the same frequencies of the diurnal and semi-diurnal ocean tides. As the radiationnal and the gravitational components cannot be well separated from observations, both components are included in global ocean tide models; thus the radiationnal tides should not be also included in the DAC correction to avoid redundancy when correcting altimetry data.

The methodology chosen to correct the operational DAC from S1 and S2 radiational tides and make it complementary to the ocean tide correction, is based on Ponte and Ray (2002); it consists in removing S1 and S2 atmospheric pressure climatologies from the DAC forcing. Climatologies computed from 11 years of operational ECMWF data (1993-2003; Carrere, 2005) are used for the operational DAC, but they are not coherent with the ERA-Interim dataset. New monthly climatologies based on 18 years of ERA-Interim pressure data (1992-2009) have been computed and then removed from the DAC pressure forcing for the present study.

Figure 2 shows the difference between the new ERA-Interim pressure climatology and the one used for operational DAC: differences are lower than 100 Pa over oceans and can be stronger on land.

### 2.3.2 The new Dynamic Atmospheric Correction derived from ERA Interim

The ERA Interim DAC correction (noted DAC_ERA) has been computed while forcing the MOG2D barotropic ocean model with the corrected ERA Interim meteorological data described above. The interest of using an atmospheric model reanalysis is to improve the quality of DAC on the oldest years and thus improve the homogeneity of the correction on the entire altimetric period; improving homogeneity helps estimating more accurate trends. Same post-processing as the one used for the reference DAC (20-days filtering) has been performed. The new correction has been computed on the 1991-2013 altimetric period.

### 2.4 The Dry Tropospheric correction

The propagation velocity of a radio pulse is slowed by dry gases and the quantity of water vapour in the Earth's troposphere. The dry gas contribution is nearly constant and produces height errors of approximately -2.3 m. This effect can be modelled as the gases in the troposphere contribute to the index of refraction. In details, the refractive index depends on pressure and temperature. When hydrostatic equilibrium and the ideal gas law are assumed, the vertically integrated range delay is a

function of the surface pressure only (Chelton, 2001). The dry meteorological  tropospheric  range correction is defined by the following formulae:

$$Dry\_Tropo = -2.277 * P_{atm}(1 + 0.0026 * \cos(2 * LAT)) \qquad\qquad (2)$$

where $P_{atm}$ is the surface atmospheric pressure in mbars, LAT  is the latitude and Dry_Tropo is the dry tropospheric correction in mm.

As there is no straightforward way of measuring the nadir surface pressure from altimetry, it is determined from a global atmospheric model. The operational dry tropospheric correction (named DT_ECMWF hereafter) is based on the ECMWF operational analyses, which have a 6-hours time resolution (cf AVISO, 2011).

### 2.4.1 Specific processing for S1 and S2 atmospheric tides

Concerning the dry tropospheric correction, the diurnal (S1) and semi-diurnal (S2) atmospheric tides also demand a specific processing because they are not well sampled by the 6-hours ECMWF pressure fields due to Nyquist theory.

The methodology chosen to correct the operational Dry_Tropo  from S1 and S2 atmospheric  tides, is based on Ponte and Ray (2002) to remove S1 and S2 atmospheric pressure climatology from the ECMWF pressure field, as described in the DAC section 3.1.1. A second step consists in adding correct S1 and S2 atmospheric tides from a specific atmospheric tide model (Ray and Ponte, 2003).

### 2.4.2 The new Dry Tropospheric Correction derived from ERA-Interim

The ERA Interim Dry Tropospheric correction (DT_ERA) is based on the ERA Interim atmospheric pressure (Mean Sea Level Pressure field), with a 6-hour temporal resolution, and specific S1S2 climatologies described in section 3.1.1. The new correction is available on the 1991-2013 period.

### 2.5 Method of comparison

In order to compare the studied corrections and to estimate their impact on the accuracy of altimeter data, the first step consists in interpolating bilinearly in space and time, the grids of DAC and DT corrections on the satellites' ground tracks. Differences between ERA-interim based corrections and the operational ECMWF ones can then be investigated along-track for each altimeter. The along-track interpolated values also allow computing the altimeter sea surface height (SSH) using successively each of the corrections, ERA-based or operational ones. The differences in the sea level contents are analyzed for different time and spatial scales. Notice that even the pressure derived corrections solely depend on the state of the atmosphere, considering several altimeters allows studying different temporal periods: for example, TP, Jason-1 and Jason-2 are consecutive datasets. Moreover as TP and ERS ground tracks have different orbit characteristics (cycle, heliosynchronism), using these two types of data allows considering different aliasing problems.

The impact of DAC_ERA and DT_ERA is primarily estimated for short temporal scales (time lags lower than 10 days), which are very significant for these corrections as they contain a large part of their variability (Vinogradova et al. 2007). Moreover these short temporal scales are indirectly linked with climate scales since high temporal frequency errors increase the formal error estimation of larger temporal scale signals.

The impact of using each of the studied corrections on the SSH performances is estimated by computing the SSH differences between ascending and descending tracks at crossovers of each altimeter, using successively the studied correction and the reference one. Crossover points with time lags shorter than 10 days within one cycle, are selected in order to minimize the contribution of the ocean variability at each crossover location. The DAC is by essence a high-frequency correction as described in section 3.1, with short temporal autocorrelation scales (Lamouroux, 2006; Mourre, 2004), and the DT is directly

proportional to pressure field (cf. section 3.2); thus this diagnostic allows a good estimation of the impact of the DAC and the DT correction on the high-frequency part of the altimeter SSH, focusing on signals with periods below 10 days in the case of this crossovers diagnostics..

The maps of the variance difference of SSH differences at crossover points using successively each altimetric components in the SSH calculation are first computed; they are computed on small boxes of 4°x4° and give information on the temporal

variance of the SSH differences. The long-term monitoring of SSH is estimated thanks to the calculation of global statistics for each altimeter cycle, all along the time span of each mission, and considering multi-missions concatenated time series as described in Fig 1; it gives information about the temporal evolution of the spatial variance of the SSH differences. For both diagnostics, the reduction of variance indicate a better internal consistency of sea-level between ascending and descending passes within a 10-days window and thus characterizes a better SSH performance. SSH differences at crossovers focuses on

HF variability and the spatial resolution of this diagnostic is limited due to the localization of crossovers.

To pursue the analysis further to the coast, we consider along-track observations instead of crossovers: the along-track SLA statistics are calculated from 1 Hz altimetric measurements. Although high frequency signals are aliased in the lower frequency band following the Nyquist theory applied to each altimeter sampling, SLA time series contain the entire ocean variability spectrum. To investigate the impact of the new DAC near the coasts, the differences of SLA variances, computed

by using successively both DAC corrections, can be plotted as a function of coastal distances between 0 and 100 km.

The analysis is finally focused on ocean long-term evolution at global and at regional scales, which is relevant for climate studies. The global and regional mean sea level (MSL) trends are computed for each altimetric mission considered here (from 1992 onwards), applying the MSL calculation method described on AVISO website: http://www.aviso.altimetry.fr/en/data/products/ocean-indicators-products/mean-sea-level.html (Ablain et al. 2011). Basically

mean grids of SLA are first computed for each cycle of each mission (every ~10 days); then the global mean of each grid is computed for each cycle to estimate the global MSL slope for each mission. The regional MSL slopes for each mission are then estimated using previous SLA grids for each cycle and each mission and a least-square method at each grid point. Trends are estimated for each SLA using successively the studied and the reference DAC and DT corrections. Notice that the trends of the altimetric missions can be very different one to each other due to the impact of the mission's timespan on the

trend estimation, longer timespan allowing a more accurate trend estimation. Error-bar of the MSL trends estimation is about 0.5 mm/yr (Ablain, 2015).

## 3. Analysis of the differences of atmospheric pressure derived corrections

In this section we analyses the differences between the reference (ECMWF based) and the studied (ERA-Interim based) atmospheric pressure derived corrections, namely DAC and DT, at a global and regional scales; a long-term analysis of these differences is also presented on the 20 years of altimetry data available.

### 3.1 The Dynamic Atmospheric Correction

The monitoring of the global differences between DAC_ERA and DAC_ECMWF corrections and also of the map of the differences provide information concerning the impact of the studied correction at the global and regional scale and for different time scales. Figure 3 shows the monitoring of the standard deviation and the mean of the differences between both corrections on 20 years period. Figure 4 shows the maps of the differences (mean and standard deviation) for the ERS-1, ERS-2 and ENVISAT missions, which cover the nearly entire altimetric period considered.

The mean difference between both corrections is about 1 mm, with annual variations below a few tenths of mm for all missions. The standard deviation of the differences clearly evolves with time, with strong differences on the first years of altimetry (until 1.6-1.8 cm for ERS-1 and TP) which decrease until year 2002 and then become stable around 0.5 cm for ENVISAT, Jason-1 and Jason-2. A low annual signal is likely explained by the seasonal ice cover's impact.

The maps of the differences also indicate stronger values for old altimeter missions: the mean of differences shows values until 1 cm or even more in some large regions mainly located in southern high latitudes for ERS-1, ERS-2, and in the Arctic and other ocean regions for ERS-1. As expected from the atmospheric pressure and wind high frequency variability, the standard deviation of differences shows weak differences in the inter-tropical area (between latitudes 40°S/40°N) and strong differences of several cm (over 3 cm) in southern high latitudes, in the Bering strait, in the Arctic and in some shallow water regions. The differences are stronger in the southern Pacific for the three old missions considered and significantly higher for the oldest one, ERS-1.

Concerning more recent missions as ENVISAT, mean differences maps show some patterns with small differences below 0.4 cm; and standard deviation maps indicate values below 0.8 cm on most of the global ocean and until 1.8-2 cm in a few shallow water regions. Those results confirm that both atmospheric models considered are very close on recent years, but some differences remain in shallow waters likely explained by the lower resolution of the reanalysis on this period.

### 3.2 The Dry Tropospheric correction

Figure 5 shows the monitoring of the global standard deviation and mean differences between DT_ERA and DT_ECMWF corrections on 20 year period. Figure 6 shows the map of the differences between both corrections (mean and standard deviation) for the ERS-1, ERS-2 and ENVISAT missions.

The mean difference between both corrections is nearly null, with variations lower than a few tenth of mm for all missions. As for the DAC, the standard deviation of the differences is stronger for old missions, with lower values than for the DAC because the DT correction is a smaller amplitude correction: differences reach 0.4 cm for TP and ERS-1 missions, and 0.1 cm for Jason-2 mission, one fourth of the differences observed for DAC. A low annual signal is also visible. We notice a small but sharp lowering of the standard deviation of differences at the beginning of year 2006; this is likely explained by the resolution change from N256 to N400 of the ECMWF native grid, and indicates that the DT correction is more affected than DAC by the meteorological model evolutions (cf. ECMWF system evolutions website).

The maps of the differences indicate also stronger values for the old missions – ERS-1 and ERS-2 - than for the more recent ENVISAT mission: the global mean differences are low for all missions (below 5 mm). Following the atmospheric pressure variability pattern, the variability of the difference is stronger in the southern high latitudes and reaches more than 1 cm for ERS-1 and until 0.7 cm for ERS-2, and only 0.2 cm for ENVISAT. We also notice some small scale oscillations on ENVISAT maps which are explained by some errors occurring in the operational DT fields based on gaussian grid of surface pressure (Gibbs oscillations) used since 2002 (Dibarboure, 2003).

### 4. Ocean short temporal scales

Analyses presented in this section concern high frequency signals (time differences lower than 10 days). In order to quantify the impact of each correction in the SSH calculation, crossovers and along-track analyses are performed as described in previous section. We first focus on the impact on global ocean and then go further into details with some regional analysis.

### 4.1 DAC

The impact of the new DAC_ERA on the SSH performance is first quantified by plotting the temporal evolution of SSH variance differences at crossovers using successively the different DAC in the SSH calculation (cf. Figure 7), respectively for the TP/Jason-1/Jason-2 and the ERS-1/ERS-2/ENVISAT altimeter time series. We note that DAC_ERA strongly reduces the SSH variance compared to the operational DAC on the first years of altimetry: the reduction reaches 5-12 cm² on the 1992-1996 period, and it corresponds to a mean diminution of the along-track SSH error of 2-3 cm when using DAC_ERA, which is a very important result. Then this impact diminishes until 2002, but it still remains significant.

Concerning more recent missions (Jason-1, Jason-2, ENVISAT), DAC_ERA and DAC_ECMWF have comparable results in terms of crossovers variance reduction: differences remain between +/- 1 cm² on average on 2002-2014. DAC_ERA tends to raise slightly the variance compared to ECMWF operational correction only from 2006 and onwards. The very close results

of DAC_ERA and DAC_ECMWF on the recent altimeter period is remarkable and not expected since the operational ECMWF model has benefited from significant improvements over time. Evolution of ECMWF operational dataset is linked to improved modeling, resolution and data assimilation process: operational database has a 0.5° resolution until 2006, to be compared to the 0.7° of ERA-Interim, then operational model resolution changed to N400 (~0.2°) in January 2006, and to N640 in 2010 (cf. ECMWF evolutions website). Global ocean results suggest that modeling and data assimilation improvements contained in ERA-Interim have a very important impact and overwhelm the lower resolution issue of ERA-Interim on most of the studied period, even until 2006. Only the last versions of ECMWF operational model tend to improve slightly DAC_ECMWF compared to DAC_ERA on the recent years.

To investigate regional patterns, the maps of SSH variance difference at crossovers using successively the DAC_ERA and the reference DAC, for each altimeter missions are plotted in Figure 8: old ones ERS-1, ERS-2, TP and recent ones Jason-1, Jason-2, EVISAT. Regionally, the improvement of sea-level estimation is very significant using the DAC solutions derived from ERA-Interim for all old missions tested, ERS-1, ERS-2 and TP: DAC_ERA allows reducing the residual variance at crossovers by more than 10 cm² in the Southern Ocean where the high frequency dynamic response of the ocean to atmospheric forcing is very important (Webb and de Cuevas 2002, 2003; Carrere, 2003; Vinogradova and Ponte, 2007). The reduction is also significant in many shallow water regions like the Bering strait, the Hudson bay, the Patagonian shelf, north Australia, in the Yellow sea…, and in the Arctic Ocean. In all those regions, DAC_ERA correction allows diminishing the along-track error by more than 3 cm, compared to DAC_ECMWF, which is very significant. Those results show that the ERA-Interim reanalysis is much more accurate than the operational ECMWF model, which is used to compute the reference DAC, on the first decade of altimetry.

If considering the second decade of altimetry (ENVISAT, Jason-1, Jason-2), both DAC have a similar impact in deep ocean regions but using DAC_ERA raises the SSH crossovers variance in some shallow water regions like the Bering strait, the Arctic ocean, the China Sea, the Patagonian Shelf or around Australia. This local variance raise can be explained by the better resolution of the operational forcing on the recent years, which is an asset to solve short spatial scales characteristic of shallow-coastal areas; this raise is stronger for ENVISAT and Jason-2 which are the most recent altimeters studied here. The impact of DAC_ERA as a function of distance to coast for global ocean is shown on Figure 9, confirming previous results: DAC_ERA allows reducing the SLA variance near the coasts for old altimeter mission while it tends to raise it slightly when considering more recent missions.

**4.2 Dry Tropospheric Correction**

The impact of the new DT_ERA on the SSH performance is evaluated thanks to the estimation of the temporal evolution of SSH variance differences at crossovers using successively different DT corrections in the SSH calculation as plotted in Figure 10. As for DAC_ERA, DT_ERA correction strongly reduces the variance compared to DT_ECMWF on the first

years of altimetry: reduction reaches 2-5 cm² on the 1992-1996 period, which corresponds to a diminution of the along-track SSH error by 1-2 cm. Even the impact of DT_ERA is weaker than DAC_ERA's one due to the smaller amplitude of the correction itself, the impact of DT_ERA is very significant on the first decade of altimetry. Then this impact diminishes until giving similar results as the operational DT_ECMWF on the 2002-2013 period. It is worth noting that the SSH variance reduction obtained with DT_ERA remains negative on the entire period, showing an improvement, even slightly on last decade; this result was not expected.

The maps of SSH variance difference at crossovers using successively DT_ERA and DT_ECMWF (cf. Figure 11) give information about the regional patterns of this improvement for each altimeter. The maximum variance reduction is localized at high latitudes, where the variability of atmospheric pressure is maximum. The regional improvement of sea-level estimation is very significant using the DT_ERA solution for all old missions, ERS-1, ERS-2 and TP: the variance gain is the strongest for ERS-1 and reaches more than 10 cm² in the high latitudes. For ERS-2 and TP, the variance gain is a bit smaller but remains significant in the southern Ocean.

If considering more recent missions (ENVISAT, Jason-1, Jason-2), we notice that DT_ERA still allows reducing the SSH variance on the global ocean, even on most recent years (Jason-2). This unexpected result is worth underlining, as the operational ECMWF pressure field benefits from a better resolution than ERA-Interim on this period which should improve the quality of the DT_ECMWF correction. However these results suggest that the impact of the Delayed-Time assimilation window used for ERA-Interim reanalysis (Dee et al., 2011) is more important than the spatial resolution for the quality of the DT correction.

## 5. Ocean long-term climate signals

The impact of using the new ERA-Interim derived atmospheric corrections (DAC_ERA and DT_ERA) instead of the operational correction is analyzed in terms of long-term trend of the altimeter SLA. Operational ECMWF analyses are known to contain drifts due to the evolution of the operational model upon time (change of computational methods and in the data assimilation system; Thorne and Vose, 2010), which can impact the MSL trend estimations (Ablain et al. 2009). As meteorological reanalyzes ensure greater homogeneity of the database over time, they are thus more suitable for long-term signals estimations as already discussed by Ablain et al. (2009) and Legeais et al. (2014); moreover, reduced high frequency errors thanks to the better quality of the reanalysis as described in previous sections, will decrease the formal error estimation of the long-term signals as the MSL trend. MSL trends at global and regional scales are investigated as described in section 2.3. Particularly as the difference between ECMWF and ERA-Interim based corrections shows large spatial patterns of strong variability, the regional MSL trends may be significantly affected by the use of the pressure derived corrections based on ERA-Interim.

Global analysis shows that the new ERA-Interim based solutions have a very small impact on the estimation of the global MSL trends of different altimeter missions considered in the study: indicates that differences of trends are smaller than 0.07 mm/yr, which is one order of magnitude lower than the global MSL trend uncertainty: 0.5 mm/yr (Ablain, 2015). Even meteorological models can have instabilities or jumps due to model evolutions, this weak impact on global trends could be expected as the mean pressure is removed to perform the IB and to force the barotropic model, and the DAC is computed with an instantaneous zero mean. As seen on Figure 5, the DT is more affected by meteorological model evolutions as it depends on the pressure field, but results indicate that the impact on MSL trend is negligible at a global scale.

Notice that the trend differences observed between each missions are not significant, as they are mostly explained by the different lengths of the temporal series available.

The impact of using ERA-Interim based corrections on the regional MSL trends estimation is analyzed in terms of spatial distribution of the MSL trends for each mission considered in the study (cf. Figure 12 and Figure 13). Although no impact is detected on the global MSL trend, using DAC_ERA correction instead of DAC_ECMWF has a significant impact on the estimation of regional trends. Considerable trend differences are displayed for the oldest missions; differences are located nearly everywhere on the global ocean for ERS-1 (±7 mm/yr) and are likely explained by the strong differences between DAC_ERA and DAC_ECMWF on this period, but also by the short time-span of the ERS-1 temporal series, which makes the MSL trends estimation less accurate and less stable. MSL trends differences are mostly located in the southern high latitudes for ERS-2 (±2.5 mm/yr) and TP (±1.5 mm/yr), which correspond to the regions where the differences between the DAC solutions themselves are the greatest and also where the SSH variance reduction is strong. Concerning more recent missions, the impact of DAC_ERA on regional MSL trends estimations is smaller than for old missions, but it is still not negligible: differences reach locally 1-1.5 mm/yr for ENVISAT, Jason-1 and Jason-2. The impact of DAC_ERA on the estimation of regional trends is likely explained by the fact that DAC_ERA strongly reduces the high frequency variability locally as discussed in previous sections, and thus the formal error of the least square adjustment of the MSL trends is also reduced; this impact is all the more important that the regional trends are more affected by the oceanic variability and annual/semi-annual signals than global trends.

Using the new DT_ERA correction instead of the DT_ECMWF has a weak impact on the regional MSL trends, as seen on Figure 13: differences are lower than 0.3 mm/yr on the global ocean for most of missions. Differences are stronger for ERS-1 mission, reaching 0.5 mm/yr or even a bit more on nearly the entire ocean, but these stronger values are likely mainly due to the shorter time series available for this mission.

The different diagnostics presented here point out some differences for long-term regional trends estimations, when using the ERA-based corrections instead of operational corrections, but they do not demonstrate which trend is the most realistic. Comparisons with in-situ measurements (Valladeau et al, 2012) as well as tide gauges or temperature and salinity profiles do not allow obtaining relevant results mainly due to the errors of the methods. However, as the DAC_ERA and DT_ERA induce strong improvements when considering short temporal scales (c.f. section 4), and because these high frequencies are

related to lower frequencies through the aliasing phenomena and contribute to the formal error estimation of longer time-scales signals, we can assume that the DAC_ERA and the DT_ERA corrections have a positive impact on regional MSL trends estimation.

## 6. Discussion and conclusions

New DAC and DT corrections derived from the ERA-Interim reanalysis have been computed on the entire altimetric period. These corrections have been extensively compared to the operational DAC and DT solutions using long time series of six altimeter missions: ERS-1, ERS-2, TP, ENVISAT, Jason-1 and Jason-2.

Concerning short temporal scales, the improvement of sea-level estimations using ERA-based corrections is maximum on first decade of altimetry due to the lower quality of operational ECMWF analysis during this period. The impact is more important at high latitudes where the atmospheric forcing is more energetic, and DAC_ERA also shows a significant improvement in shallow waters where the ocean has a strong dynamic response to atmospheric forcing at high frequencies. Using the new DAC_ERA correction induces a diminution of the along-track SSH error of about 1-2.4 cm globally and even more than 3 cm at high latitudes and in shallow waters. Although the DT correction has a lower variability compared to the DAC, using the new DT_ERA allows reducing the along-track SSH error by 1-2 cm on global ocean and by more than 3 cm at high latitudes.

Unexpectedly on the three recent missions studied (ENVISAT, Jason-1 and Jason-2), ERA-based corrections show similar performances as the operational corrections although the meteorological reanalysis has a larger spatial resolution than the ECMWF operational analyses. Moreover DT_ERA remains better then DT_ECMWF on the global ocean even on the most recent mission Jason-2. DAC_ERA and DAC_ECMWF have comparable results in deep ocean, but DAC_ERA tends to raise the residual crossovers variance in some shallow water regions, where the finer resolution of operational forcing seems more appropriated to solve small spatial scales characteristic of shallow and coastal ocean dynamic.

Concerning long temporal scales relative to climate studies, the present analysis shows that the ERA-based corrections do not have a significant impact on the global MSL trends. Using DT_ERA does not impact the regional MSL trends either. Using DAC_ERA has a strong effect on long-term regional trends estimation, with trend differences of several mm/yr locally.

As the DAC_ERA induces a strong improvement when considering short temporal scales, and because these high frequencies are related to lower frequencies through the aliasing phenomena and contribute to the formal error of the MSL trends estimation, we can assume that the DAC_ERA has a positive impact on regional MSL trends estimation.

The results presented here allow recommending the use of DAC_ERA on first altimetry decade for ERS-1, ERS-2, and TP missions. For more recent missions, DAC_ERA can also be used at least for long-term signals estimation to get rid of any

discontinuity between both DAC corrections, but at the cost of a slightly raised variance in some shallow water regions. Indeed, if using a combination of DAC_ERA and operational DAC, the continuity between both DAC solutions at regional scales will need to be checked at least for long-term studies.

The Dry Troposphere correction derived from ERA-Interim pressure field is also of great interest for all applications, and

this correction can be used for all altimeter missions even the most recent one studied here, Jason-2.

Given the results of the present study, the DAC_ERA and the DT_ERA time series are still being completed in delayed-time with a few months delay. These ERA-based corrections are used in several projects and products like REAPER (2014), CCI-phase-2 project (Ablain et al, 2015), SALP (SSALTO/DUACS 2015), FES2012 and FES2014 tidal models (Carrere et al.

2012; 2014), Jason-1 reprocessing project (Jason-1 handbook, 2015) …

As the ERA-Interim meteorological product has a coarse spatial resolution compared to the operational database on recent years, a perspective of this work will be to test a new atmospheric climatology with a finer spatial grid; this would likely help improving the results presented here in shallow waters and also in the southern deep ocean regions where the ocean response to meteorological forcing is enhanced on topography patterns.

**Acknowledgements**

This work has been performed within the framework of the ESA Climate Change Initiative (CCI) and the SALP (CNES) projects. We thank Paul Poli from ECMWF for the elements of discussion he gave us on the ERA-Interim reanalysis.

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

**Table 1: Cycles used for the analysis of each altimeter mission.**

| Mission | Cycles used | Period |
| --- | --- | --- |
| Topex/Poseidon | 11-481 | 31/12/1992-08/10/2005 |
| Jason-1 | 1-330 | 15/01/2002-26/12/2010 |
| Jason-2 | 1-200 | 12/07/2008-16/12/2013 |

| | | |
|---|---|---|
| ERS-1 | 15-27 and 41-53 | 23/10/1992-02/06/1996 |
| ERS-2 | 1-85 | 15/05/1995-02/07/2003 |
| ENVISAT | 10-93 | 30/09/2002-18/10/2010 |

**Table 2: Impact of ERA-Interim based corrections (DAC_ERA and DT_ERA) on global MSL trends and the Least Square Root estimation error (LSR) in mm/yr**

| Altimeter mission | MSL trend using ECMWF corrections (Reference) +/- LSR (mm/yr) | Difference of MSL trend: ECMWF - DAC_ERA | Difference of MSL trend: ECMWF - DT_ERA |
|---|---|---|---|
| ERS-1 | 6.34 +/- 0.62 | 0.07 | 0.01 |
| ERS-2 | 2.66 +/- 0.15 | 0.01 | -0.02 |
| TP | 3.12 +/- 0.03 | -0.02 | 0.01 |
| EN | 2.28 +/- 0.18 | -0.04 | -0.03 |
| J1 | 2.55 +/- 0.07 | 0 | -0.02 |
| J2 | 3.18 +/- 0.15 | 0.07 | 0.07 |

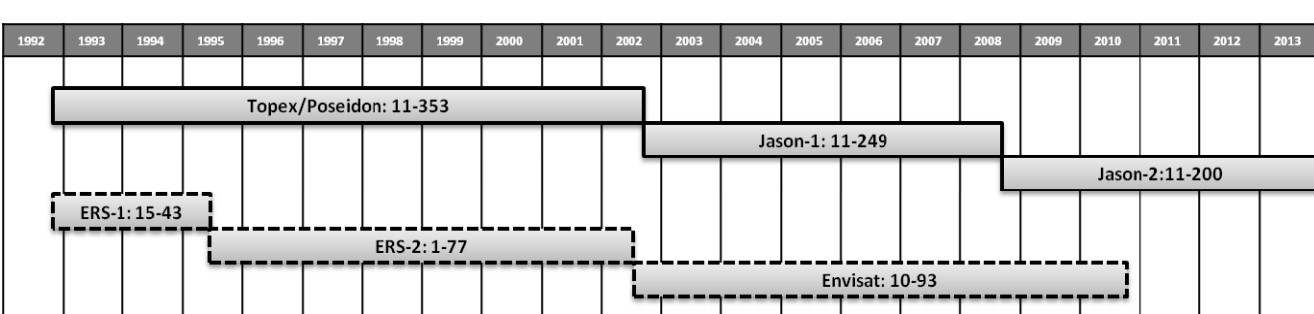

**Figure 1: Altimeter long-term time series used in the study.**

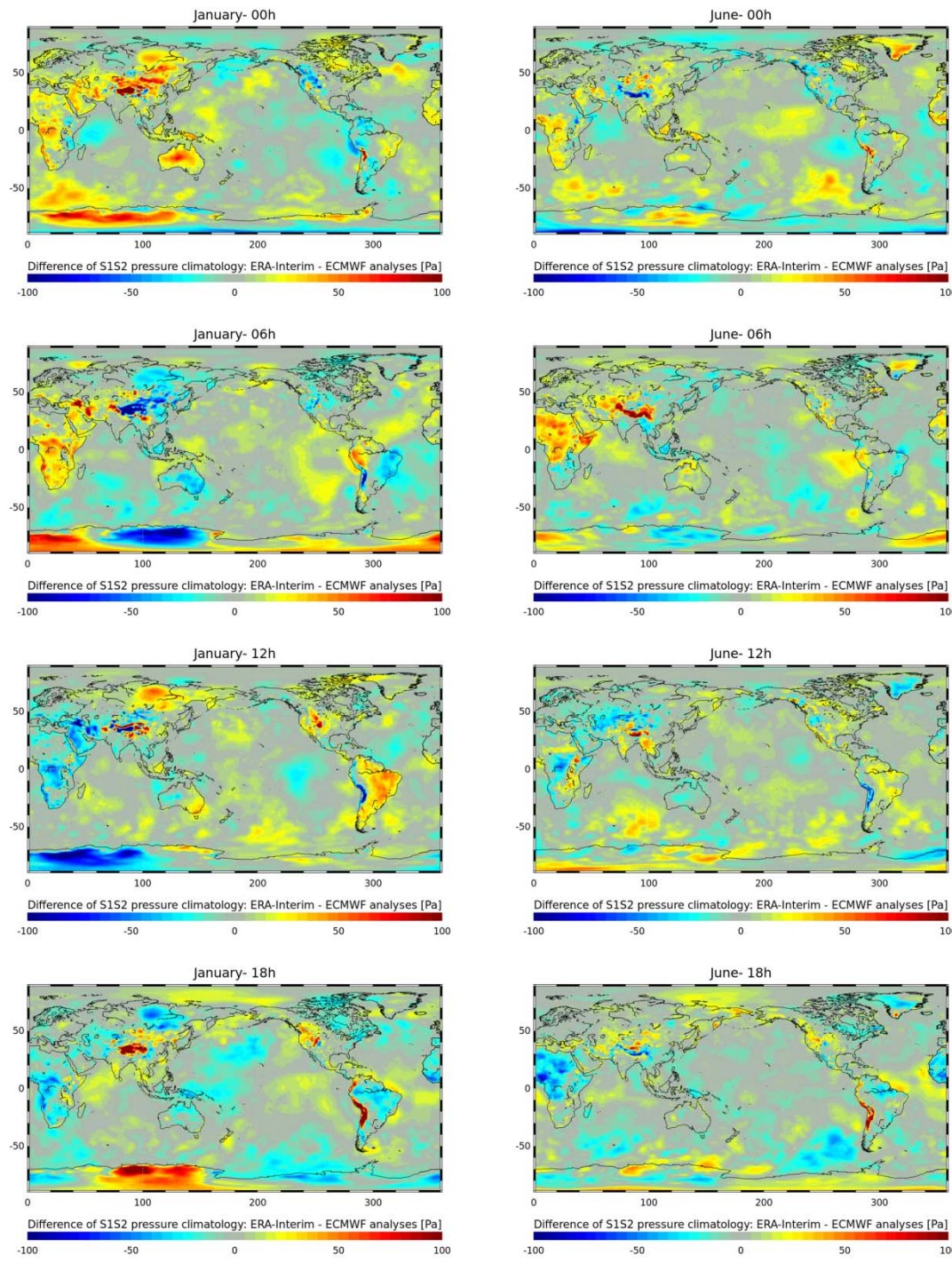

**Figure 2: Difference of S1S2 atmospheric pressure climatologies from ERA-Interim and ECMWF analyses, where S1 and S2 respectively represent the diurnal and semi-diurnal atmospheric tides.**

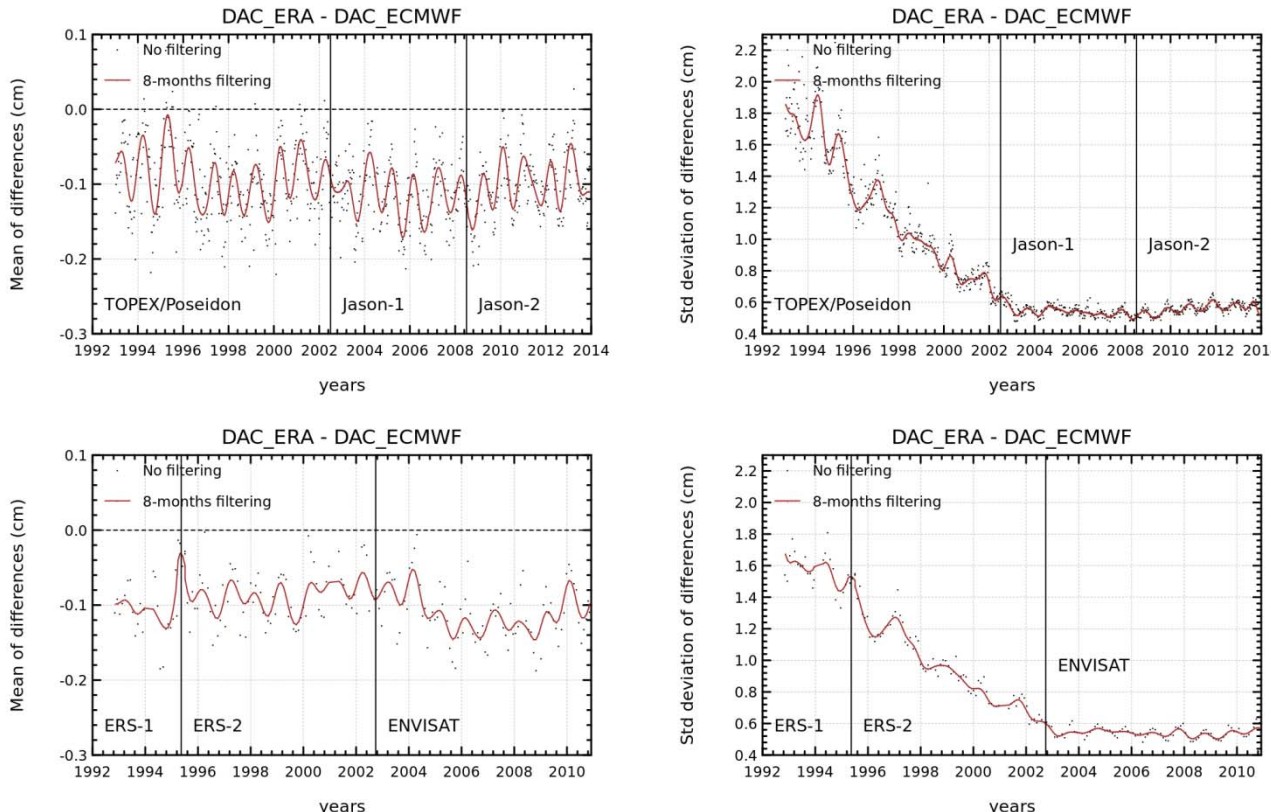

**Figure 3: Temporal evolution of the global differences between DAC_ERA and the operational DAC seen by each altimeter missions : TP, Jason-1 and Jason-2 above and, ERS-1, ERS-2, Envisat below (mean and standard deviation in cm).**

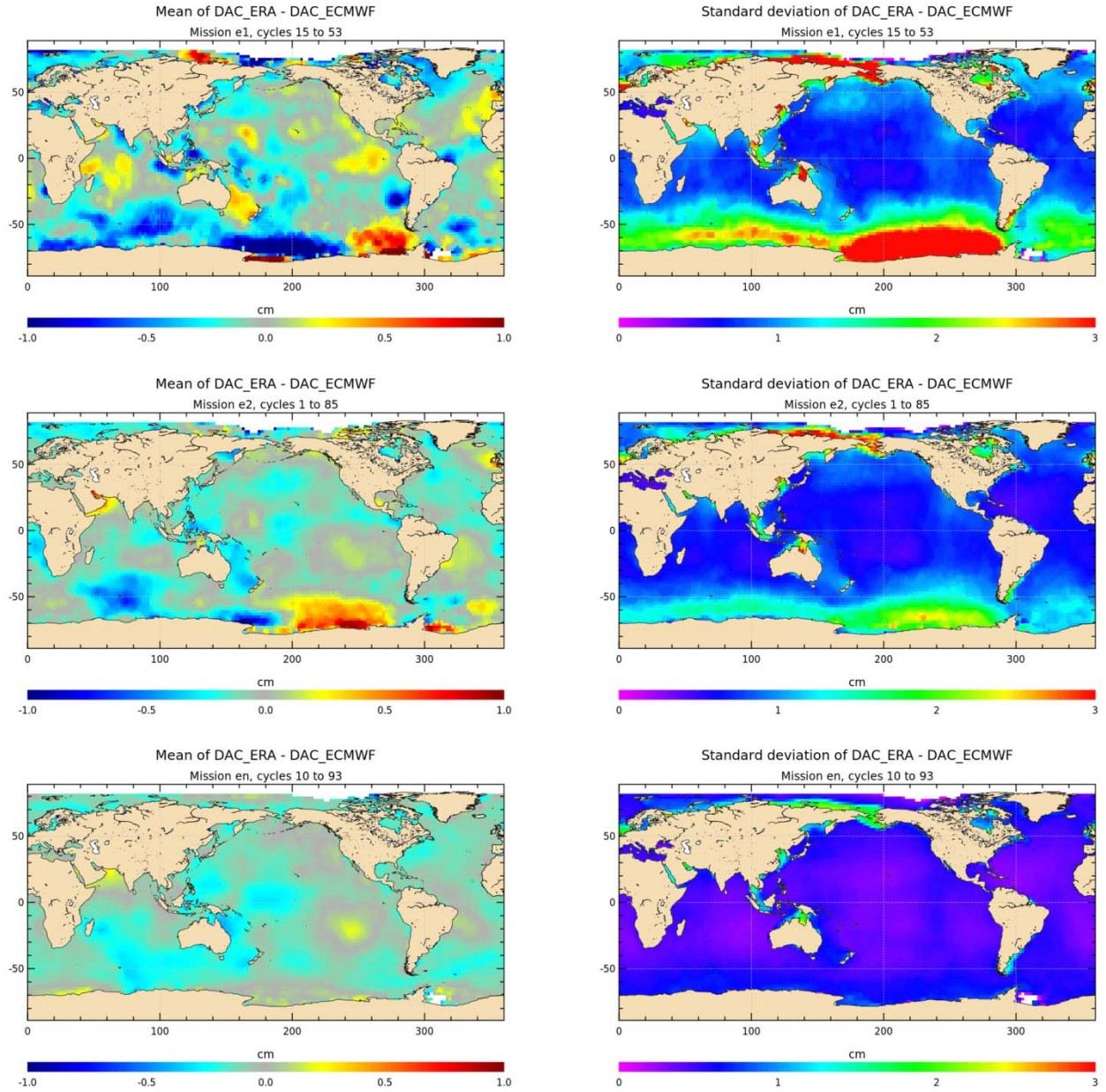

**Figure 4: Statistics of differences between DAC_ERA and the operational DAC seen by ERS-1, ERS-2 and Envisat altimeter missions (mean and standard deviation in cm).**

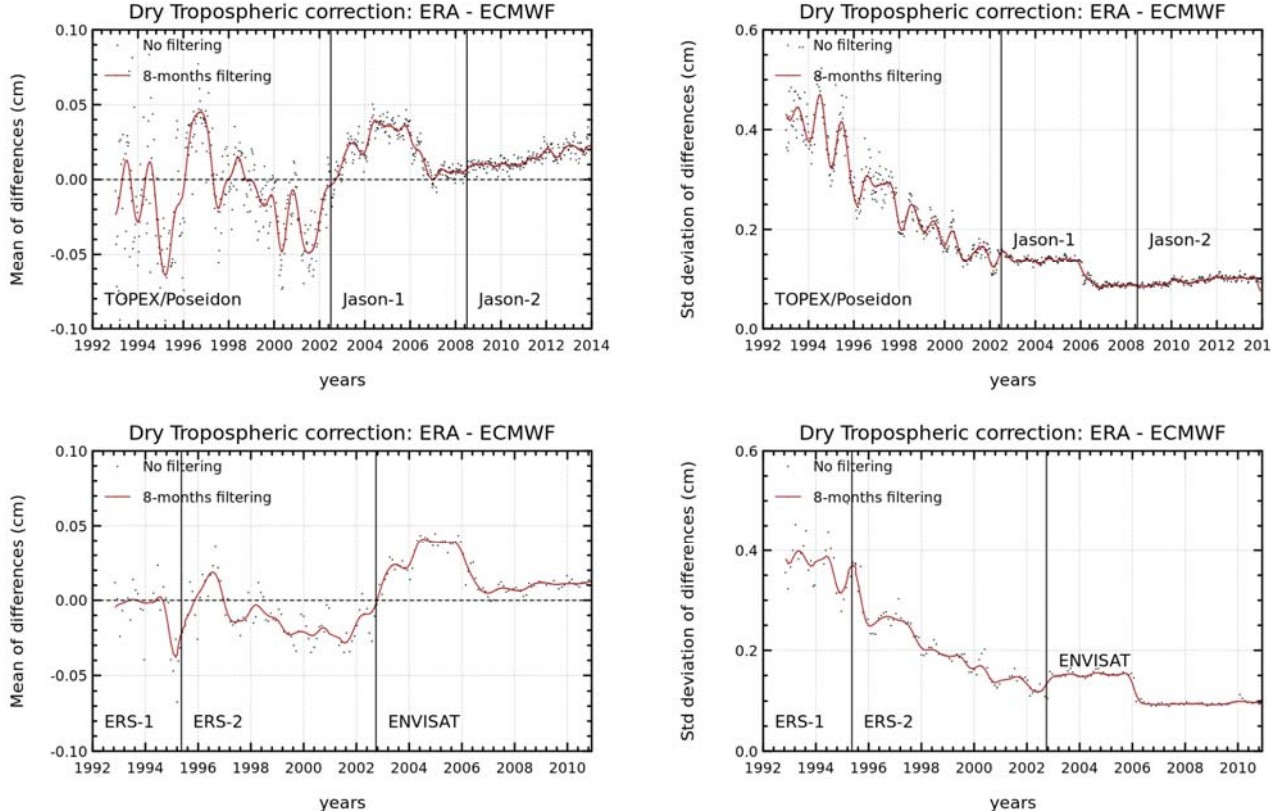

**Figure 5: Temporal evolution of the differences between Dry-Tropo-ERA and the operational Dry Tropospheric Correction seen by each altimeter missions series : TP, J1, J2 time series above, and ERS-1, ERS-2, Envisat time series below (mean and standard deviation in cm).**

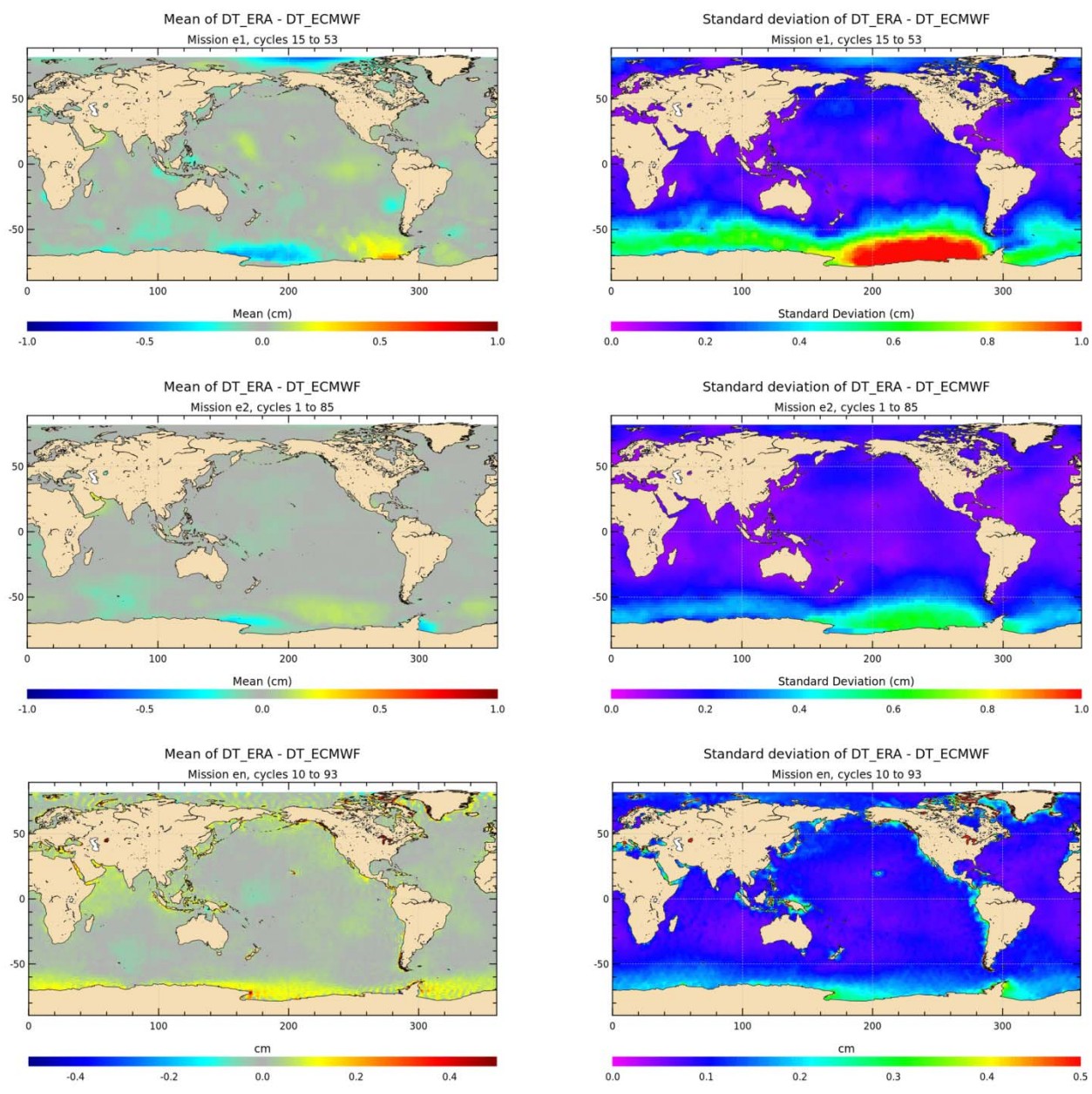

**Figure 6 : Maps of differences between DT_ERA and the ECMWF operational DT seen by altimeter missions ERS-1, ERS-2 and Envisat (mean and standard deviation in cm).**

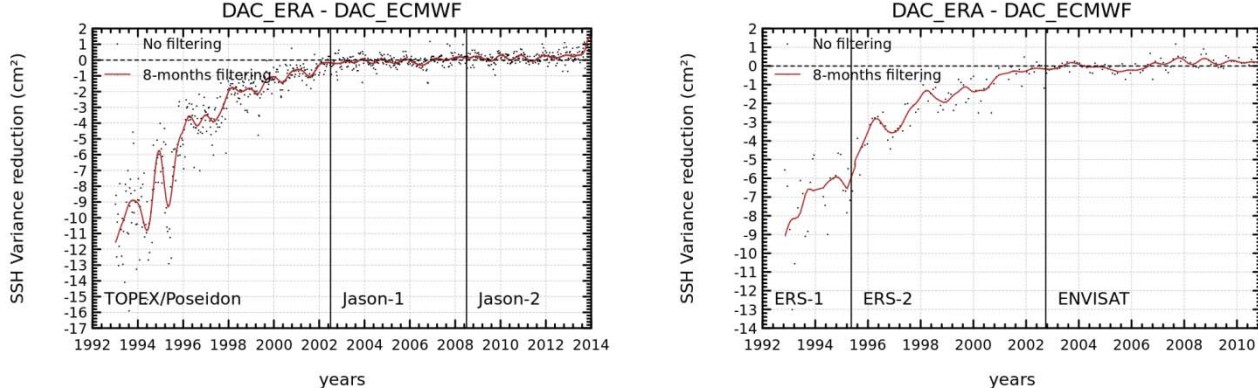

**Figure 7: Temporal evolution of SSH variance differences at crossovers using successively the ERA-Interim and reference DAC solutions in the SSH calculation for TOPEX/Jason-1/Jason-2 time series (on left), and ERS-1/ERS-2/Envisat time series (on right).**

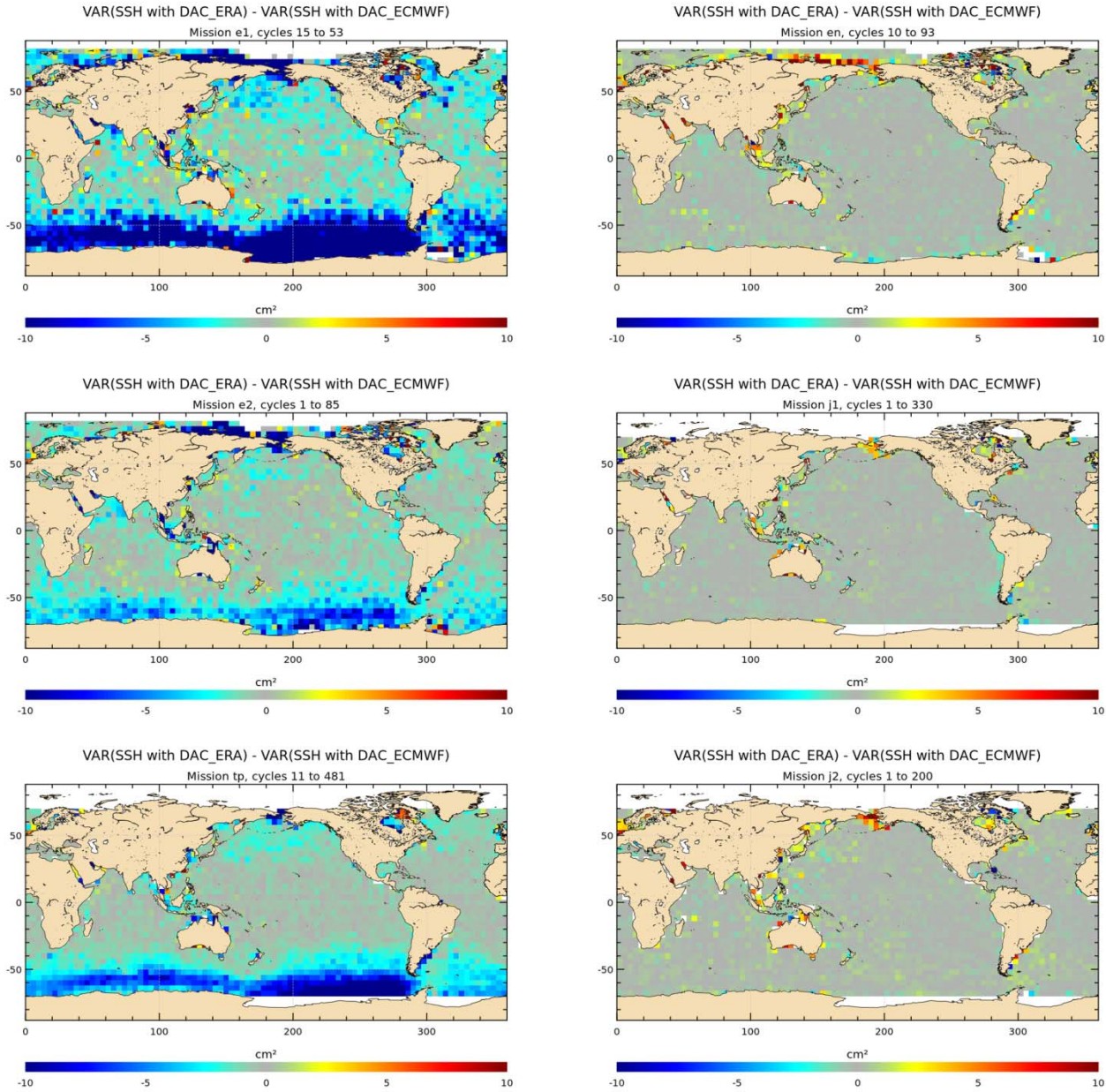

**Figure 8: Maps of SSH variance differences at crossovers using successively the ERA-Interim and reference DAC solutions in the SSH calculation for ERS-1, ERS-2 and TP on left, and for Envisat, J1 and J2 on right panel (cm²).**

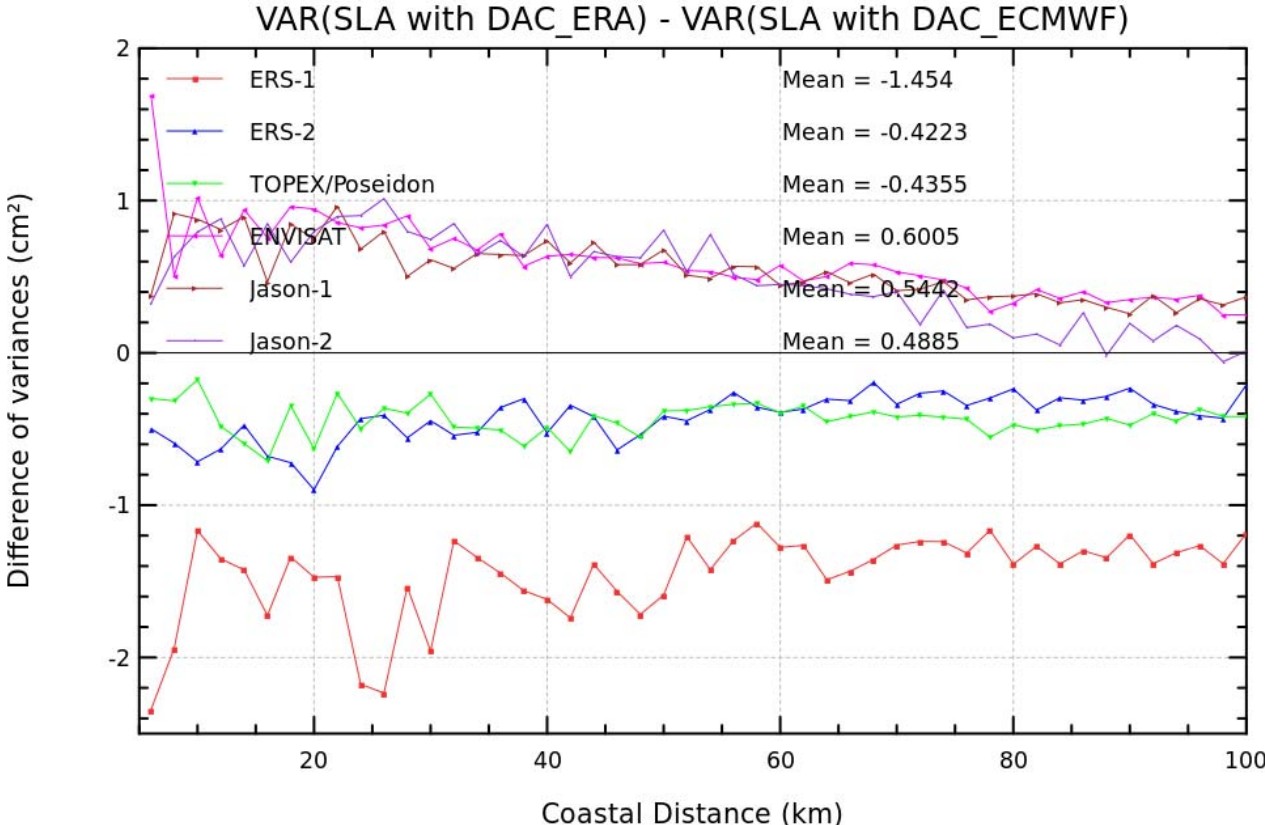

**Figure 9: Difference of variance of SLA using successively the ERA-Interim and reference DAC solutions in the SSH calculation, for each altimeter, and as a function of distance to coast.**

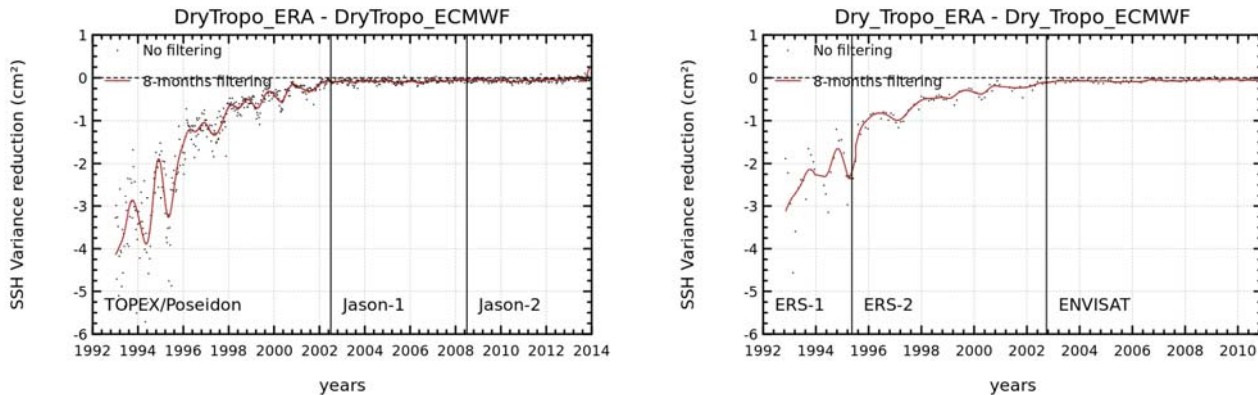

**Figure 10: Temporal evolution of SSH variance differences at crossovers using successively the ERA-Interim and ECWMF operational DT corrections in the SSH calculation for TOPEX/Jason-1/Jason-2 series (on left), and ERS-1/ERS-2/Envisat (on right).**

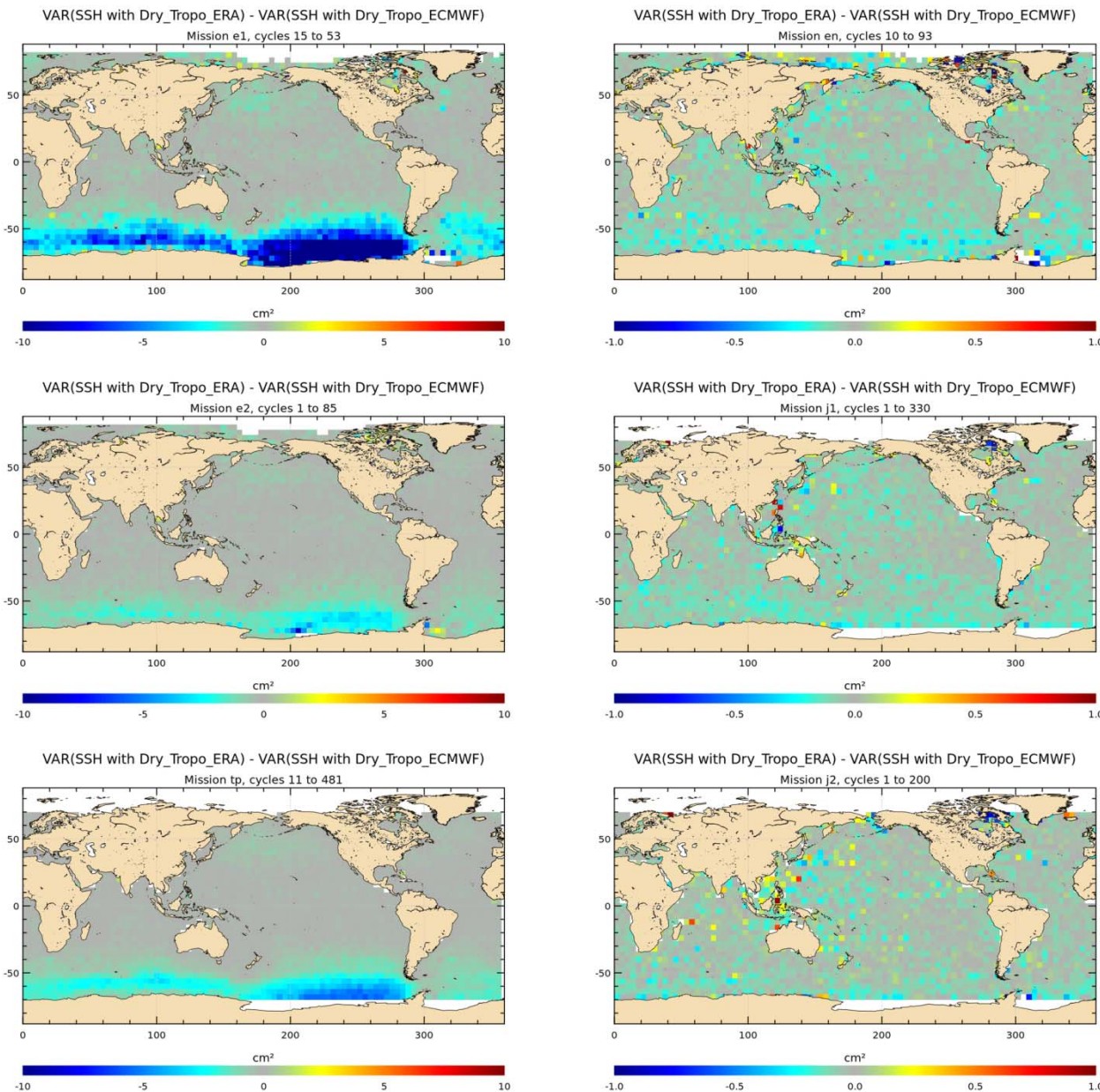

**Figure 11: Maps of SSH variance differences at crossovers using successively the ERA-Interim and ECMWF operational DT corrections in the SSH calculation for ERS-1, ERS-2 and TOPEX on left, and for Envisat, Jason-1 and Jason-2 on right panel (cm²).**

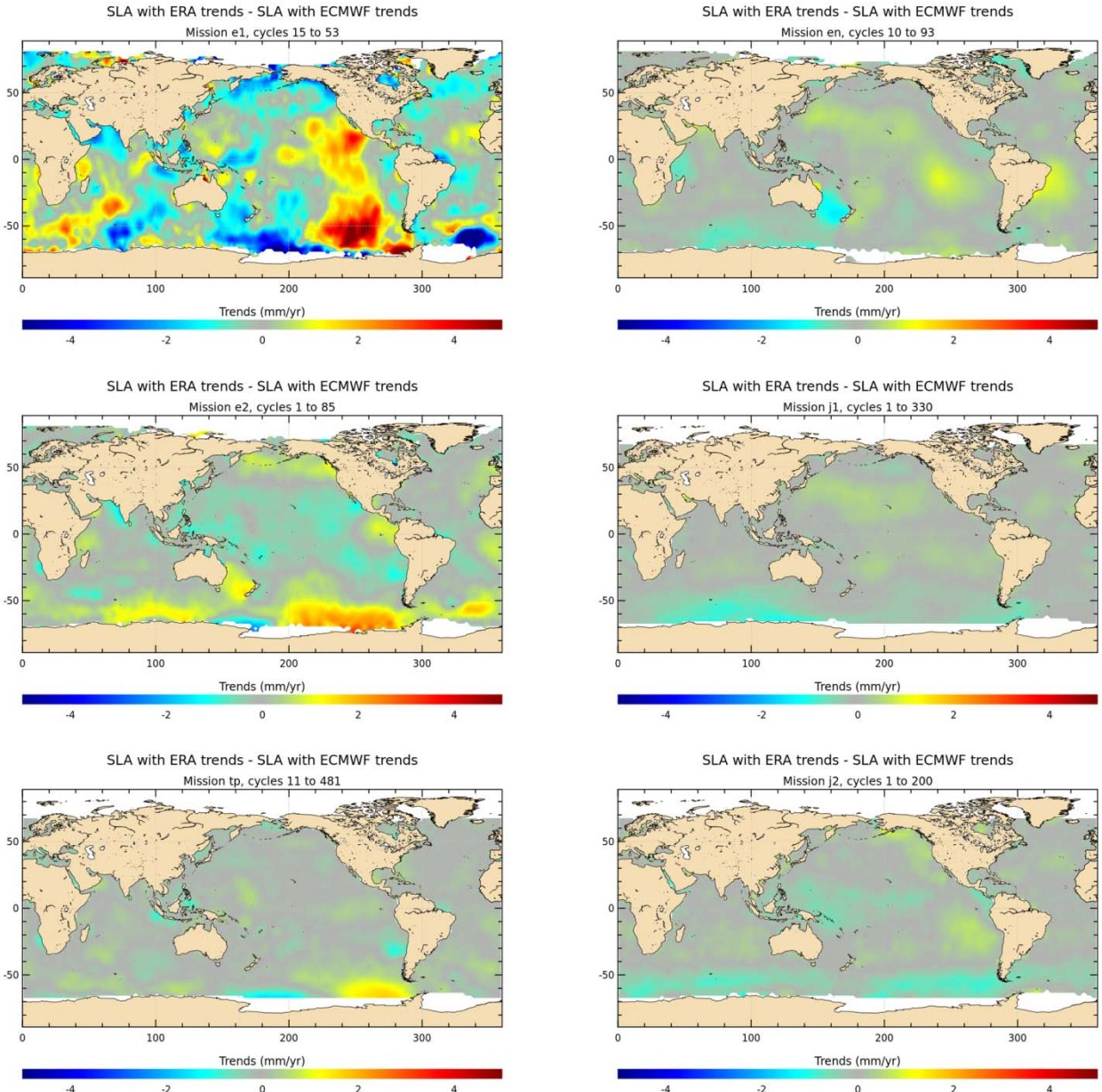

**Figure 12: Maps of MSL trend differences using successively the DAC derived from ERA-Interim and from ECMWF operational pressures fields (reference) for ERS-1, ERS-2 and TOPEX on left-hand panels, for Envisat, Jason-1 and Jason-2 on right-hand panel (mm/year).**

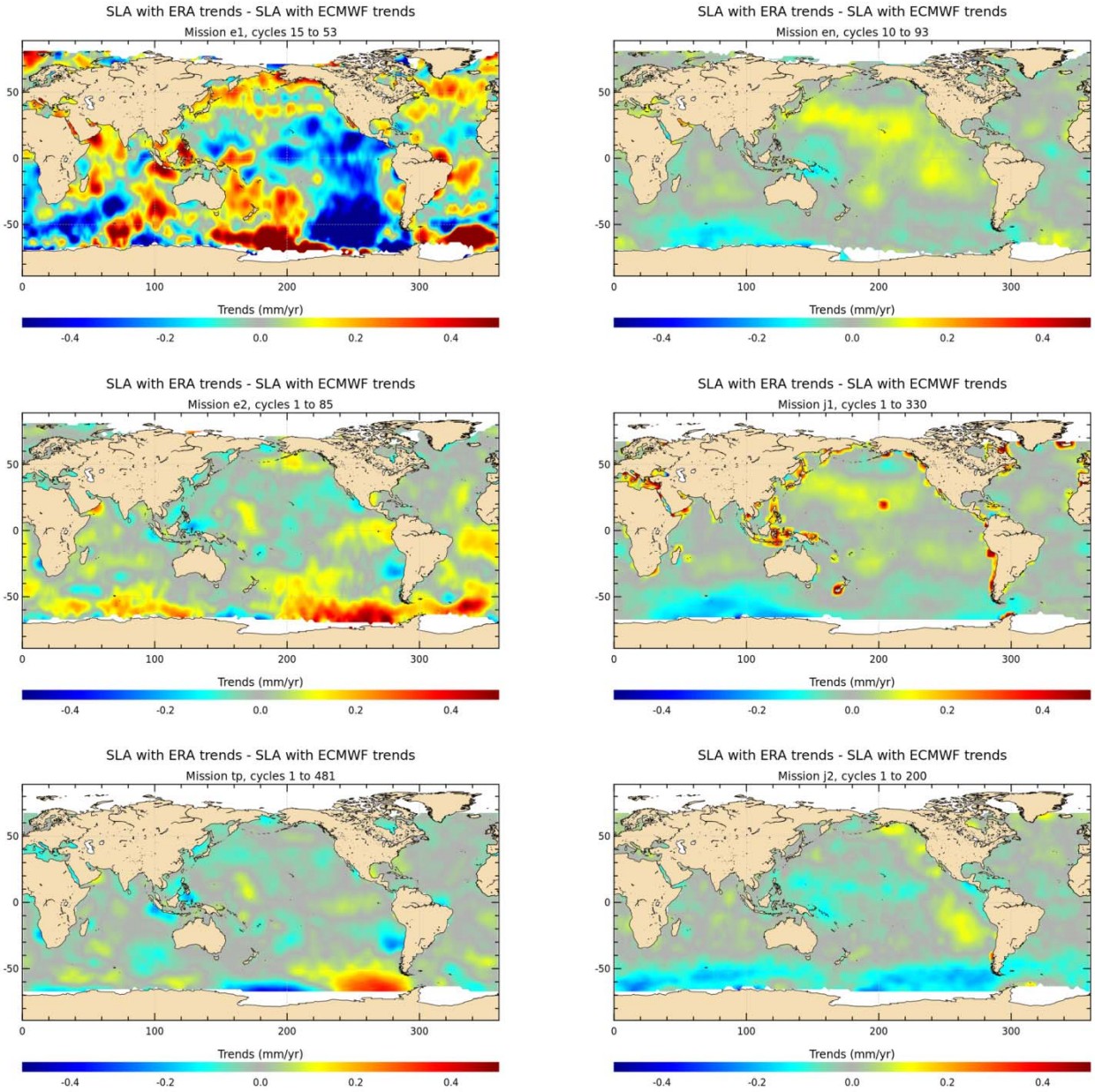

**Figure 13: Maps of MSL trend differences using successively the DT correction derived from ERA-Interim and from ECMWF operational pressures fields (reference) for ERS-1, ERS-2 and TOPEX on left-hand panels, for Envisat, Jason-1 and Jason-2 on right-hand panel (mm/year).**

