# Peer review of "Major improvement of altimetry sea level estimations using pressure derived corrections based on ERA-Interim atmospheric reanalysis"

_Ocean Science, 2015_

## Referee Comment (RC1) · L. Sun (Referee) · 1 Mar 2016

General comments: The authors claimed a major progress with a diminution of the along-track SSH error of about 2-3 cm globally and even more than 3 cm at high latitudes and in shallow waters by using the new DAC_ERA correction. This is a notable progress for short-term and/or high frequent ocean dynamics, especially for the ocean mesoscale eddies. It could be accepted for publication after some major reversions.

Specific comments: The major progress is only mentioned in conclusion, but in the abstract. The abstract is vague in addressing their main improvements and contributions. The authors may want to summarize their results and make some more comprehensive conclusions.

[Figure]

There are too many abbreviations, which should be explained at the first time. The authors may also want to list them in a table. The authors must attempt to present their work in a logical fashion.

Pg 6,line 29 what are the "trends", are they really become "more accurate"? The authors addressed long-term trends in section 5, where "results indicate that the impact on MSL trend is negligible at a global scale".

Pg 7, section 3.1.2, The authors may want give the formula of DAC correction.

Pg12, section 6, Can the authors make some comments on the feature progress on the short-term SLA products, given high resolution of ERA-based atmospheric reanalysis and altimeter missions?

Some examples are:

Pg 1, ERA-interim (Line 1, Line 12) or ERA-Interim (line 14) ?

Pg 2, Line 24, what the "IB" is? I find the answer at Pg 5, line 31!

Pg 3, Line 29 and Pg 5, line 6-7. The definition of SLA should be more consistent.

Pg 4, Line 29, the high-frequency part -> the high-frequency (noted HF)

Pg 5, Line 13, mean sea level (MSL)-> MSL

Pg 5, Line 23, The high frequency (noted HF) -> The HF. . .

Pg 5, line 25 TP (e.g. Pg 4, L17) or T/P (e.g. Pg 5, L25) ? Similar for "J1 and Jason-1","J2 and Jason-2", "DAC-ECMWF" -> "DAC_ECMWF"

Pg 9, line 10, 1996-1992 -> 1992-1996

Pg 10, line 16, 5-2 cm$^2$ -> 2-5 cm$^2$

Tables, The title of table should be above on the table.

Table 2, "ECMWF - DAC-ERA" should be "ECMWF – DAC_ERA", and similar for "DT-

ERA".

Figure 2, where S1 and S2 respectively represents the diurnal and semi-diurnal atmospheric tides.

Figure 3, "Left and right" should be "above and bellow". "temporal" ->"Temporal".

Figure 4, "statistics" ->"Statistics".

Figure 5, "temporal" ->"Temporal".

The references should be reedited. I find that the information (e.g., pages, paper id) in references is incomplete.
* * *

---

## Referee Comment (RC2) · Anonymous Referee #2 · 15 Mar 2016

General comments:

The manuscript addresses the effect on satellite altimetry estimates of the height of the sea surface resulting from the application of new geophysical corrections using reanalysis (ERA-interim) pressure values instead of the ECMWF-based operational corrections. I think that the issue is relevant and the manuscript is scientifically sound. I recommend publication after some revision, mostly minor and concerning mainly the clear presentation of the work rather than the content itself. In this line, some improvement on the use of the English language would also be beneficial.

I would suggest some restructuring of the presentation to improve the readability and clarity of the paper. For example, the DAC is only introduced in detail in page 5 (section

3.1), and then it is explained that it comes from a barotropic model forced by pressure and wind, but in the introduction the issue with the ECMWF used in the barotropic model is already stated (pg 2, line 22). As another example, the grids of DAC and DT corrections are referred in section 2.2 (pg 4, line 12) without any hint on how those grids are derived from the pressure values until later in the paper. Although a reader familiar with satellite altimetry processing can follow the ideas, I think that the presentation could be improved to make the paper easier to understand for a wider oceanographic readership. The acronyms should be fixed in the beginning and kept as simple as possible – for example, why use Dry_Tropo_ECMWF or DT_ECMWF (pg 7, line 25)?

The comparison of Table 1 and Figure 1 can be confusing because of the differing start cycles used for the individual missions and the long-term series. It should be clear in the text whether the long term series correspond to spatially averaged observations (global MSL) or along-tracks time series resulting from the concatenation of observations from different missions.

Specific comments:

Page 4, line 30: "is calculated from a cyclic way..." - perhaps rephrase (not clear)

Page 5, line 3: "To go further to the coast..." - improve presentation, in the present formulation it appears that to go further to the coast SLA is considered instead of SSH, while what is meant is I think that to approach the coast along-track observations are considered instead of crossovers

Page 8, line 5: section 0

Page 10, line 28: SSH anomalies... SLA? (keep notation consistent)

Page 12, line 9: c.f.

Page 13, line 21: ...

Page 13,line 22: "misses a smaller spatial resolution"... is not clear, I suggest rephrase

Figure 3: harmonize titles of upper and lower plots

Figure 10: the figure for TP/J1/J2 (left) seems to show a jump in SSH variance reduction at the end of 2013. Is it an artifact of the filter used, or a real feature?

Table 2: please add the corresponding uncertainties to the values presented in Table 2 (e.g 0.07 +- . . . ). Is there a reason for the differences being 0 for J1 and highest for J2?

[Figure]

---

## Referee Comment (RC3) · R. Scharroo (Referee) · 21 Mar 2016

I am very pleased to have read this paper, as it makes very clear the progress made in determining the DAC for altimetry using the ERA-Interim sea level pressure and wind speed fields.

Nonetheless, I have numerous suggestions to improve the paper. Some are merely textual, some conceptual, but none ought to be too hard to apply in a minor revision.

Comments below are indicated by a pair of (page/line number).

(2/8) "deeply improved" -> "greatly improved"

[Figure]

(2/11) "IB correction" was not previously introduced/explained.

(4/5) "If compared" -> "When compared"

(4/7) "improved compared" -> "better than"

(4/8) "Six-hour ERA-interim analysis" -> "Six-hourly ERA-interim analysis grids"

(4/24) "each studied" -> "each of the studied"

(4/26) "10 days". Why not 5 days, as this is the maximum time difference between ascending and descending tracks. With 10-day time differences you will create duplicate statistics, using the same measurements more than once.

(4/30) "HF part". "HF" was not previously introduced or explained.

(4/32) "computed ;" -> "computed;"

(6/1-6/8) Why would different filtering be used for TOPEX/Jason on the one hand (20 days) and ERS/Envisat on the other (70 days)? DAC is a physical phenomenon independent on the sampling frequency (just as tides). Of course, there are different aliasing characteristics just like tides, but the DAC per se is the physical phenomenon of the sea level variation due to pressure and wind ("weather") just like tides are due to sun and moon. Also, the Nyquist frequency is not 20 days, but 10 days in crossovers. So why the insistence on using a 20-day filter. Please explain.

(6/17) "radiationnal" -> "radiational"

(6/24) "Ocean ... Earth." -> "oceans ... land."

(7/4) "EN" -> "Envisat" (as in most other cases)

(8/1) "altimeter" -> "altimetry"

(8/7-8/10) Would it not be better to "remove-replace", i.e. remove the S1-S2 atmospheric tides from the model grids, then replace it along-track.

**OSD**

(9/9) "cm" -> "cmˆ2"

(9/10) "2-3 cm" -> "1-2 cm". I doubt this range. First: the difference DAC_ERA-DAC_ECMWF is no more than 1.8 cm std.dev. globally, so how can the SSH error reduce 2-3 cm std.dev. globally? Second: the crossover variance reduction is quoted as 5-12 cmˆ2, which leads to SSH variance reduction (on a single track) of 2.5-6 cmˆ2, which suggests an error reduction of 1.6-2.4 cm maximum. Therefore 1-2 cm is a better indication.

(9/13) "+/- 1 cm" -> "+/- 1 cmˆ2"

(9/15 and a number of other places) "DAC-ECMWF" -> "DAC_ECMWF"

(10/26) "ERA-1" -> "ERS-1"

(10/27) "10 cm" -> "10 cmˆ2"

(11/30) "nearly the global" -> "nearly everywhere across the global"

(12/29) "2-3 cm" -> "1-2 cm" (see comment on (9/10))

---

## Author Comment (AC1) · 10 May 2016

Dear L. Sun,

Thank you for your review and comments. My answers and suggestions for changes are noted in blue in the text below.
Best regards
Loren Carrere

General comments: The authors claimed a major progress with a diminution of the along-track SSH error of about 2-3 cm globally and even more than 3 cm at high latitudes and in shallow waters by using the new DAC_ERA correction. This is a notable progress for short-term and/or high frequent ocean dynamics, especially for the ocean mesoscale eddies. It could be accepted for publication after some major reversions.

Specific comments: The major progress is only mentioned in conclusion, but in the abstract. The abstract is vague in addressing their main improvements and contributions. The authors may want to summarize their results and make some more comprehensive conclusions.

LC: The abstract has been reformulated to show more clearly the results of the present paper. Conclusions have been slightly changed

There are too many abbreviations, which should be explained at the first time. The authors may also want to list them in a table. The authors must attempt to present their work in a logical fashion.

LC: OK all abbreviations have been homogenized and defined when first used in the first pages of the paper.
The beginning of the paper has been a bit modified to propose a more logic approach, clearer for the reader:
We propose to move the definition of DAC and DT in part 2, and change part 3 into 'differences of atmospheric pressure derived correction'.
New plan is thus, included moved sections noted en bold :

15

Pg 6,line 29 what are the "trends", are they really become "more accurate"? The authors addressed long-term trends in section 5, where "results indicate that the impact on MSL trend is negligible at a global scale".

LC: the impact of the non- homogeneity of the pressure forcing (particularly jumps in the time series) on the MSL trend

20 estimation has been addressed in Ablain et al. 2009 (for inverse barometer and DT components). In the case of the DAC, this impact is indeed negligible for global MSL trend but it is significant at regional scale. This impact is explained by the fact that using the new DAC_ERA improves the high-frequencies and thus it diminishes a component of the estimation error.

Pg 7, section 3.1.2, The authors may want give the formula of DAC correction.

25 LC: OK the formula of DAC correction has been added in section 3.1-The dynamic atmospheric correction. The equations of the DAC, which are the shallow waters equations, are given in previous papers (Carrere and Lyard 2003; Carrere 2003).

Pg12, section 6, Can the authors make some comments on the feature progress on the short-term SLA products, given high resolution of ERA-based atmospheric reanalysis

30 and altimeter missions?

LC: The main interest of ERA-interim is its constant resolution on the time series, the constant model version used and the more numerous assimilated datasets. Its slightly higher spatial resolution compared to the operational model on first years of altimetry, combined to other improvements mentioned above, help to have a better localization and a more realistic

amplitude of high/low pressure systems, which directly impacts the quality of the pressure derived corrections (DAC and DT).

If considering short-temporal scales, using DAC_ERA impacts large-scale features which characterize the high frequencies ocean response to atmospheric forcing: several thousands of km in deep ocean and a few hundreds km in shallow waters due to non-linearities (Vinogradova et al, 2007;Webb and de Cuevas, 2002). Using DT_ERA impacts large spatial scales as it is directly dependant from surface pressure.

Temporal and spatial scales impacted are similar from one altimeter mission to the other, as the DAC is defined by a unique temporal filtering as described in section 3.1; however as the time spans of the missions are different (length and period), and the cycles are also different implying different aliasing frequencies, the validation statistics can vary from one mission to the other.

Some examples are:

Pg 1, ERA-interim (Line 1, Line 12) or ERA-Interim (line 14) ?

LC: OK notation has been homogenized to ERA-Interim in the entire paper.

Pg 2, Line 24, what the "IB" is? I find the answer at Pg 5, line 31!

LC: IB is Inverse Barometer. Definition has been added on page 2.

Pg 3, Line 29 and Pg 5, line 6-7. The definition of SLA should be more consistent.

LC: OK correction has been made.

Pg 4, Line 29, the high-frequency part -> the high-frequency (noted HF)

LC: OK text has been corrected.

Pg 5, Line 13, mean sea level (MSL)-> MSL

LC: OK text has been changed.

Pg 5, Line 23, The high frequency (noted HF) -> The HF

LC: OK text has been changed.

Pg 5, line 25 TP (e.g. Pg 4, L17) or T/P (e.g. Pg 5, L25) ? Similar for "J1 and Jason-1","J2 and Jason-2", "DAC-ECMWF" -> "DAC_ECMWF"

LC: OK notations have been homogenized in the entire paper.

Pg 9, line 10, 1996-1992 -> 1992-1996

LC: OK text has been changed

Pg 10, line 16, 5-2 cm2 -> 2-5 cm2

5  LC: OK text has been changed

Tables, The title of table should be above on the table.

LC: OK title put above the tables

10  Table 2, "ECMWF - DAC-ERA" should be "ECMWF – DAC_ERA", and similar for "DT_ERA".

LC: OK text has been changed.

Figure 2, where S1 and S2 respectively represents the diurnal and semi-diurnal atmospheric
tides.

15  LC: OK text has been added in the legend of the figure.

Figure 3, "Left and right" should be "above and bellow". "temporal" ->"Temporal".

LC: OK text has been added in the legend of the figure

20  Figure 4, "statistics" ->"Statistics".

LC: OK text has been added in the legend of the figure

Figure 5, "temporal" ->"Temporal".

LC: OK text has been added in the legend of the figure

25

The references should be reedited. I find that the information (e.g., pages, paper id) in
references is incomplete.

LC: OK references have been checked and completed.

30

---

## Author Comment (AC2) · 10 May 2016

Dear Referee,

Thank you for your review and comments. My answers and suggestions for changes are noted in blue in the text below.

Best regards

Loren Carrere

General comments:

The manuscript addresses the effect on satellite altimetry estimates of the height of the sea surface resulting from the application of new geophysical corrections using reanalysis (ERA-interim) pressure values instead of the ECMWF-based operational corrections. I think that the issue is relevant and the manuscript is scientifically sound. I recommend publication after some revision, mostly minor and concerning mainly the clear presentation of the work rather than the content itself. In this line, some improvement on the use of the English language would also be beneficial.

I would suggest some restructuring of the presentation to improve the readability and clarity of the paper. For example, the DAC is only introduced in detail in page 5 (section3.1), and then it is explained that it comes from a barotropic model forced by pressure and wind, but in the introduction the issue with the ECMWF used in the barotropic model is already stated (pg 2, line 22). As another example, the grids of DAC and DT corrections are referred in section 2.2 (pg 4, line 12) without any hint on how those grids are derived from the pressure values until later in the paper. Although a reader familiar with satellite altimetry processing can follow the ideas, I think that the presentation could be improved to make the paper easier to understand for a wider oceanographic readership.

LC: The beginning of the paper has been a bit modified to propose a more logicapproach, clearer for the reader:

We propose to move the definition of DAC and DT in part 2, and change part 3 into 'differences of atmospheric pressure derived correction'.

New plan is thus, included moved sections noted en bold :

The acronyms should be fixed in the beginning and kept as simple as possible – for example, why use Dry_Tropo_ECMWF or DT_ECMWF (pg 7, line 25)?

LC: OK acronyms have homogenized in the all paper.

The comparison of Table 1 and Figure 1 can be confusing because of the differing start cycles used for the individual missions and the long-term series. It should be clear in the text whether the long term series correspond to spatially averaged observations (global MSL) or along-tracks time series resulting from the concatenation of observations from different missions.

LC: Figure 1 has been modified because the starting cycle of ERS-1 was wrong: it is cycle 15 as in table 1. The long-term series are either the along-track (fig 3, fig 5) or the crossovers (fig 7, fig 10) time series resulting from the concatenation of the cycles from different missions (TP-J1-J2 on one side and ERS1-ERS2-ENVISAT on the other side, cf figure 1). For global MSL estimations (table 2), mean grids of SLA are first computed for each cycle of each mission (every ~10 days as described on AVISO web site); then the global mean of each grid is computed for each cycle to estimate the  MSL slope. The regional MSL slopes for each mission are then estimated using previous SLA grids for each cycle and each mission and a least-square method at each grid point.

    ⇨   Text has been clarified in section  2.3.

Specific comments:

Page 4, line 30: "is calculated from a cyclic way..." - perhaps rephrase (not clear)

LC: OK sentence has been rephrased => 'The long-term monitoring of SSH is estimated thanks to the calculation of statistics for each altimeter cycle, all along the time span of each mission; …'

Page 5, line 3: "To go further to the coast..." - improve presentation, in the present formulation it appears that to go further to the coast SLA is considered instead of SSH, while what is meant is I think that to approach the coast along-track observations are considered instead of crossovers

LC: OK the sentence has bee rephrase to be clearer : 'To pursue the analysis further to the coast, we consider along-track observations instead of crossovers…'

Page 8, line 5: section 0

LC: put explicit reference to DAC = section 3.1.1 (= section 2.3 in new organization proposed)

Page 10, line 28: SSH anomalies: : : SLA? (keep notation consistent)

LC: p11 line 5 => SLA

Page 12, line 9: c.f.

LC: OK changed

Page 13, line 21: …

LC: I don't understand this comment.

Page 13,line 22: "misses a smaller spatial resolution": : : is not clear, I suggest rephrase

LC: OK rephrased : "As Era-Interim … has a coarse spatial resolution compared to the operational database on recent years, …"

Figure 3: harmonize titles of upper and lower plots

LC: OK title have been harmonized.

Figure 10: the figure for TP/J1/J2 (left) seems to show a jump in SSH variance reduction at the end of 2013. Is it an artifact of the filter used, or a real feature?

LC: This is an artifact of the filter used.

Table 2: please add the corresponding uncertainties to the values presented in Table
2 (e.g 0.07 +- : : : ). Is there a reason for the differences being 0 for J1 and highest for
J2?

LC : The Least Square Root Error has been added in Table 2 (column 2); note that this theoretical adjustment error is
underestimated as observations are not decorrelated in reality.

The differences observed between J2 and J1 are due to the different temporal series of the missions: J2 time series is shorter
than for J1, which implies a higher estimation error for J2 (0.15 mm/yr instead of 0.07 mm/yr for J1) and thus likely stronger
differences when using different corrections. The fact the mean impact of DAC_ERA  is 0 for J1 is just that the value is <
0.01 so it was round off to 0. For DT_ERA the impact on J1 is -0.2 mm/yr and not 0.

---

## Author Comment (AC3) · 10 May 2016

Dear Remko Scharroo,

Thank you for your review and comments. My answers and suggestions for changes are noted in blue in the text below.
Best regards
Loren Carrere

I am very pleased to have read this paper, as it makes very clear the progress made in determining the DAC for altimetry using the ERA-Interim sea level pressure and wind speed fields.

Nonetheless, I have numerous suggestions to improve the paper. Some are merely textual, some conceptual, but none ought to be too hard to apply in a minor revision.

Comments below are indicated by a pair of (page/line number).

(2/8) "deeply improved" -> "greatly improved"

LC: OK text has been changed

(2/11) "IB correction" was not previously introduced/explained.

LC: OK. Replaced by : "a static inverse barometer correction (IB)".

(4/5) "If compared" -> "When compared"

LC: OK text has been changed.

(4/7) "improved compared" -> "better than"

LC: OK text has bee changed.

(4/8) "Six-hour ERA-interim analysis" -> "Six-hourly ERA-interim analysis grids"

LC: OK text has been changed.

(4/24) "each studied" -> "each of the studied"

LC: OK text has been changed

(4/26) "10 days". Why not 5 days, as this is the maximum time difference between ascending and descending tracks. With 10-day time differences you will create duplicate

statistics, using the same measurements more than once.

LC: Indeed this 10-day time difference is to be used with all missions (TPJ, ERS-EN …) and it aims at minimizing the impact of the oceanic variability on the crossovers differences, while keeping enough points for statistical analysis. This threshold of 10 days is used for each cycle separately thus avoiding using same measurement more than once even for TPJ..

5

(4/30) "HF part". "HF" was not previously introduced or explained.

LC: OK text has been clarified..

(4/32) "computed ;" -> "computed;"

10   LC: OK text has been changed.

(6/1-6/8) Why would different filtering be used for TOPEX/Jason on the one hand (20 days) and ERS/Envisat on the other (70 days)? DAC is a physical phenomenon independent on the sampling frequency (just as tides). Of course, there are different

15   aliasing characteristics just like tides, but the DAC per se is the physical phenomenon of the sea level variation due to pressure and wind ("weather") just like tides are due to sun and moon. Also, the Nyquist frequency is not 20 days, but 10 days in crossovers. So why the insistence on using a 20-day filter. Please explain.

LC: I agree that DAC contains a physical phenomenon (response of the ocean to wind and pressure forcing) as described in

20   p5/l 22-28, but for altimetry the main purpose of this correction is to remove the HF ocean variability (due to wind and pressure forcing) which is aliased in the lower frequency band due to bad temporal sampling of altimeters. Thus talking of aliasing we could envision using different filtering wavelengths for the different missions, but the choice was to keep the same filtering for all the missions with a wavelength based on the reference mission aliasing.

I also agree that the Nyquist frequency of crossovers is shorter compared to along-track, but the DAC is applied on along-

25   track products and it is thus optimized for these along-track products and aliasing problems. Moreover 10 days crossovers are mainly used for validation as described in section 2.3. But if you would like to use crossovers time series, thus benefitting form the higher temporal sampling of crossovers, you could also use a DAC with a 10-day filter. For the moment only one DAC is provided in altimeter products, but it can be envisioned to provide a non filtered DAC for users who would like to optimize the filtering wavelength with their specific analysis/purposes…

30

(6/17) "radiationnal" -> "radiational"

LC: OK text has been changed.

(6/24) "Ocean ... Earth." -> "oceans ... land."

LC: OK text has been changed.

(7/4) "EN" -> "Envisat" (as in most other cases)

LC: OK text has been changed.

(8/1) "altimeter" -> "altimetry"

LC: OK text has been changed.

5 (8/7-8/10) Would it not be better to "remove-replace", i.e. remove the S1-S2 atmospheric

tides from the model grids, then replace it along-track.

LC: indeed a remove-replace process is done along-track: S1 and S2 aliased atmospheric tides are removed using gridded S1S2 pressure climatologies (Ponte and Ray 2002), and they are then replaced by correct S1 and S2 atmospheric tides using a specific atmospheric tide model (Ray and Ponte 2003). => Text has been changed to be clearer.

10 (9/9) "cm" -> "cm^2"

LC: OK. cm² already noted in the text but there is likely an edition problem with ² => text has been changed

(9/10) "2-3 cm" -> "1-2 cm". I doubt this range. First: the difference DAC_ERA-DAC_

ECMWF is no more than 1.8 cm std.dev. globally, so how can the SSH error

reduce 2-3 cm std.dev. globally? Second: the crossover variance reduction is quoted

15 as 5-12 cm^2, which leads to SSH variance reduction (on a single track) of 2.5-6 cm^2,

which suggests an error reduction of 1.6-2.4 cm maximum. Therefore 1-2 cm is a better

indication.

LC: You are right, I missed a 2 factor between crossovers and SSH. "2-3 cm" is replaced by "1-2.4 cm" in the text.

(9/13) "+/- 1 cm" -> "+/- 1 cm^2"

20 LC: OK. cm² already noted in the text but there is likely an edition problem with ² => text has been changed

(9/15 and a number of other places) "DAC-ECMWF" -> "DAC_ECMWF"

LC: OK name has been homogenized

(10/26) "ERA-1" -> "ERS-1"

LC: OK text has been changed.

25 (10/27) "10 cm" -> "10 cm^2"

LC: OK. cm² already noted in the text but there is likely an edition problem with ² => text has been changed

(11/30) "nearly the global" -> "nearly everywhere across the global"

LC: OK text has been changed.

(12/29) "2-3 cm" -> "1-2 cm" (see comment on (9/10))

30 LC: OK "2-3 cm" is replaced by "1-2.4 cm" in the text.

---

## Author Response (AR1)

[revised manuscript text omitted]

[Figure]

**Figure 3:** Temporal evolution of the global differences between DAC_ERA and the operational DAC seen by   each altimeter missions :  TP, Jason-1 and Jason-2 above and, ERS-1, ERS-2, Envisat below (mean and standard deviation in cm).

[Figure]

**Figure 4: Ss**tatistics of differences between DAC_ERA and the operational DAC seen by ERS-1, ERS-2 and Envisat altimeter missions (mean and standard deviation in cm).

[Figure]

**Figure 5: T**emporal evolution of the differences between Dry-Tropo-ERA and the operational Dry Tropospheric Correction seen by each altimeter missions series : TP, J1, J2 time series above, and ERS-1, ERS-2, Envisat time series below (mean and standard deviation in cm).

[Figure]

**Figure 6 : Maps of differences between DT_ERA and the ECMWF operational DT seen by altimeter missions ERS-1, ERS-2 and Envisat (mean and standard deviation in cm).**

[Figure]

[Figure]

**Figure 7: Temporal evolution of SSH variance differences at crossovers using successively the ERA-Interim and reference DAC solutions in the SSH calculation for TOPEX/Jason-1/Jason-2 time series (on left), and ERS-1/ERS-2/Envisat time series (on right).**

[Figure]

**Figure 8: Maps of SSH variance differences at crossovers using successively the ERA-Interim and reference DAC solutions in the SSH calculation for ERS-1, ERS-2 and TP on left, and for Envisat, J1 and J2 on right panel (cm²).**

[Figure]

Figure 9: Difference of variance of SLA using successively the ERA-Interim and reference DAC solutions in the SSH calculation, for each altimeter, and as a function of distance to coast.

[Figure]

[Figure]

**Figure 10: Temporal evolution of SSH variance differences at crossovers using successively the** ERA-Interim **and ECWMF operational DT corrections in the SSH calculation for TOPEX/Jason-1/Jason-2 series (on left), and ERS-1/ERS-2/Envisat (on right).**

[Figure]

**Figure 11: Maps of SSH variance differences at crossovers using successively the ERA-Interim and ECMWF operational DT corrections in the SSH calculation for ERS-1, ERS-2 and TOPEX on left, and for Envisat, Jason-1 and Jason-2 on right panel (cm²).**

[Figure]

**Figure 12: Maps of MSL trend differences using successively the DAC derived from ERA-Interim and from ECMWF operational pressures fields (reference) for ERS-1, ERS-2 and TOPEX on left-hand panels, for Envisat, Jason-1 and Jason-2 on right-hand panel (mm/year).**

[Figure]

**Figure 13: Maps of MSL trend differences using successively the DT correction derived from ERA-Interim and from ECMWF operational pressures fields (reference) for ERS-1, ERS-2 and TOPEX on left-hand panels, for Envisat, Jason-1 and Jason-2 on right-hand panel (mm/year).**